# Stabilized D₂R G protein-coupled receptor oligomers identify multi-state β-arrestin complexes

Katie L. Sharrocks [1,2], Francesca Fanelli[3], Yewei Liu [1], Annabelle J. Milner [2], Wu Yining[1], Bernadette Byrne [1] ✉ & Aylin C. Hanyaloglu [2] ✉

The G protein-coupled receptor (GPCR) superfamily directs central roles in many physiological and pathophysiological processes via diverse and complex mechanisms. GPCRs can exhibit signal pleiotropy via formation of di/oligomers both with themselves and other GPCRs. A deeper understanding of the molecular basis and functional role of oligomerization would facilitate rational design of activity-selective ligands. A structural model of the D2 dopamine receptor (D₂R) homomer identified distinct combinations of substitutions likely to stabilize protomer interactions. Molecular modelling of β-arrestin-2 (βarr2) bound to predicted dimer models suggests a 2:2 receptor: βarr2 stoichiometry, with the dimer favouring βarr2 over Gαi coupling. A combination of biochemical, biophysical and super-resolution, single molecule imaging approaches demonstrated that the D₂R mutant homomers exhibited greater stability. The mutant D₂R homomers also exhibited bias towards recruitment of the GPCR adaptor protein βarr2 with either faster or ligand-independent βarr2 recruitment, increased internalization and reprogrammed regulation of ERK signaling. Through GPCR dimer-stabilization, we propose that D₂R di/oligomerization has a role in βarr2-biased signaling.

G protein-coupled receptors (GPCRs) are the largest family of proteins in mammals, expressed in a range of cell types, and regulate a diverse set of physiological processes[1]. They are the targets of approximately 35% of current FDA approved drugs and remain major drug discovery targets[2,3]. GPCRs can regulate several intracellular signaling cascades through coupling with distinct heterotrimeric G-proteins. In addition, GPCRs can diversify their signaling through interactions with the multifunctional adaptor proteins, the non-visual arrestins; β-arrestin-1 and 2 (βarr1 and βarr2, respectively, also known as arrestins 2 and 3)[4,5]. Canonically, βarr1/2 are recruited to activated, GPCR kinase (GRK) phosphorylated receptors to uncouple GPCRs from their heterotrimeric G-proteins, drive receptor internalization via clathrin-coated pits, and mediate distinct cellular pathways through activation of signaling such as the mitogen-activated protein kinase (MAPK) pathway[4,5].

Class A "rhodopsin-like" GPCRs can function as monomers; however, increasing biophysical, computational, pharmacological, and physiological evidence suggests that they can form both homomers and heteromers with other GPCRs[6–12]. A dynamic equilibrium between monomers and transient di/oligomers has been reported for several GPCRs[13–16]. Oligomerization can impact the function of receptors, altering ligand efficacy, G-protein activation, or preference for G-protein coupling selectivity[17–20]. The impact of oligomerization on GPCR/β-arr function is less clear but a recent study demonstrated that oligomerization of platelet-activating factor receptor (PAFR) decreases βarr recruitment and, conversely, βarr seemed to decrease oligomerization of the receptor[21]. Additionally, βarr has been proposed to play a role in stabilizing agonist-induced μ-opioid receptor dimers[22]. β₂AR has also been shown to dimerize in a βarr-dependent manner upon

[1]Department of Life Sciences, Imperial College London, London, UK. [2]Department of Metabolism, Digestion and Reproduction, Imperial College London, London, UK. [3]Department of Life Sciences, University of Modena and Reggio Emilia, Modena, Italy. ✉e-mail: b.byrne@imperial.ac.uk; a.hanyaloglu@imperial.ac.uk

stimulation with a βarr-biased ligand[23]. This highlights the link between receptor oligomerization and signaling, with the potential for oligomerization to dramatically influence all aspects of GPCR signaling.

Structural biology remains an important area for uncovering GPCR activation and dimerization mechanisms[24], yet there have been limited reports of high-resolution structures of Class-A GPCR dimers/oligomers. Transmembrane domains often play a key role in dimer interfaces, with the most recurrent interfaces involving helix 1 (H1). Indeed, H1-H1 contacts were found in homo-dimeric rhodopsin (PDB: 2I36, 2I37, 3CAP, and 6OFJ), κ-opioid (PDB: 4DJH), β1-adrenergic (PDB: 4GPO), M3-muscarinic (PDB: 5ZHP), EP3-prostanoid (PDB: 6AK3), AT1-angiotensin (PDB: 6DO1), and apelin (PDBs 6WOL and 6WON) receptors. Furthermore, H1-H4 contacts were found in homo-dimeric 5HT2A-serotonin (PDBs: 6WGT and 6WH4) and CCR5-chemokine (PDBs: 4MBS and 6AKX) receptors, whereas H1-H5 contacts were found in homo-dimeric D4 dopamine receptor (PDB: 6IQL). Homo-dimers characterized by H4-H5 or H4-H6 H5-H5, or H6-H7 were found for CXCR4, β1-AR, C5a1, CCR2, α2C-AR and CysLT1 receptors (PDBs: 3ODU/3OE8/3OE9, 5F8U, 5O9H, 6GPX, 6KUW, 6RZ5, respectively).

The $D_2$ dopamine ($D_2R$) is a Class A "rhodopsin-like" GPCR, coupling to $G\alpha_{i/o}$[25]. The primary physiological roles of $D_2R$ in the brain are regulation of locomotion and in reward and reinforcement mechanisms[26], while pathophysiologically this receptor is implicated in a number of neurological diseases, including schizophrenia and Parkinson's Disease[27]. $D_2R$ has been shown to form homodimers both in heterologous expression systems at physiological expression levels and in native tissues[14,28–30]. Several antipsychotics target the $D_2R$/ βarr2 complex and an increase in $D_2R$ homodimers has been associated with schizophrenia[31,32]. Thus, $D_2R$ complexes represent excellent therapeutic targets. While there are several high-resolution structures of monomeric $D_2R$, also in complex with G-protein[33–35], the lack of information on precise molecular interactions between protomers and the absence of high-resolution dimer structures has hindered the ability to specifically target these complexes for therapeutic benefits.

Here, we employ a convergent strategy to engineer distinct stabilized $D_2R$ homodimers. Residues are identified within the H1-H2 interface to enhance homodimer/oligomer stability and assessed biochemically, biophysically, and via super-resolution single molecule approaches. Interestingly, these engineered receptors also exhibit a bias towards βarr2 associated pathways with either altered temporal kinetics of ligand-mediated complexes or functionally biased ligand-independent GPCR-arrestin complexes. Docking of βarr2 to the predicted homodimers of wild type (WT) or mutated $D_2R$ provide insights into the signaling active supramolecular assembly. This may highlight a selective role for $D_2R$ homomers in promoting distinct transitions to βarr2 association and signaling activity.

## Results

### $D_2R$ mutations at the H2-H2 interface increase $D_2R$ homodimer stability and protomer proximity

To stabilize a $D_2R$ homodimer, our experimental design was guided by molecular modeling (see Methods). Simulations of $D_2R$ homodimerization were carried out using the structural models of either the inactive or the active states of the receptor. The results shown here concern symmetrical $D_2R$ homodimers with both protomers in their active states (i.e., extracted from the 8IRS PDB structural complex between ritigotine-bound $D_2R$ and heterotrimeric Gαi), since they were most reliable in terms of membrane-topology indices. The predicted homodimer forms a symmetrical interface characterized by contacts at the extracellular ends of H1-H1, H1-H2 and H2-H2 and H1-H8, H8-H8 contacts on the cytosolic side (hereafter referred to as the H1-H2 dimer) (Fig. 1a). The stronger contacts from both protomers in the extracellular half of the interface involve: (a) Y36 (H1) and F102 (extracellular loop 1 (EL1)); (b) T39 (H1) and W90 (H2) and; (c) T42 (H1) and I45 (H1) (d) V97 (H2) from both protomers; and (e) V96 and Y93 (both in H2). The same

interface was found for symmetric $D_2R$ homodimers with both protomers in the inactive state (i.e., extracted from the 6CM4 PDB structural complex of $D_2R$ bound to risperidone) but not for asymmetric inactive-active $D_2R$ dimers. The presence of at least one protomer in the inactive state produced an additional interface characterized, among others, by H4-H4 contacts. For the H1-H2 dimer, we focused on the most extracellular amino acid residues to potentially drive formation of disulfide bridges. To strengthen the contacts between protomers at the interface, our models suggested mutations of Y93 to cysteine (Y93C), V96 to either serine (V96S) or cysteine (V96C), and V97 to cysteine (V97C) alone or in combination with V96S (V96S/V97C).

To assess the effect of these mutations on levels of di/oligomeric protein present in cells, Western blot analysis under reducing or non-reducing conditions (in the presence or absence of β-mercaptoethanol) was carried out. Under non-reducing conditions receptors were primarily in a higher order, dimeric/oligomeric, complex (>180 kDa) (Supplementary Fig. 1). Under reducing conditions, the majority of the WT $D_2R$ shifted in size predominately to a monomeric form (predicted size of FLAG-tagged $D_2R$-Rluc8, -90 kDa with glycosylated $D_2R$-Rluc8 appearing at -110 kDa), however, a greater proportion of V96C, V96S and in particular, V96S/V97C mutants, existed in a higher order complex (Fig. 1b, c). Given the inability to distinguish between different types of higher-order oligomeric complexes using SDS-PAGE, quantification of only relative WT and mutant $D_2R$ dimer proteins levels was compared (Fig. 1b, c). To directly measure receptor-receptor interactions in intact cells, bioluminescence resonance energy transfer (BRET) saturation assays were carried out. The WT $D_2R$ exhibited a saturation profile indicating formation of homomers (Fig. 1e–g and Supplementary Fig. 2a and 3a), as previously reported by BRET[28]. When compared to WT $D_2R$, a significant increase in BRETmax in cells expressing $D_2R$ V96C, V96S, or V96S/V97C (Fig. 1e–m), indicating an increase in protomer proximity within $D_2R$ homomers. No significant changes in the receptor interaction profile were observed for Y93C (Supplementary Fig. 2b–d). All receptors exhibited a similar total expression level (Supplementary Fig. 2e–h). This supports the data obtained by Western blot analysis that the V96S/V97C mutation in particular may stabilize and/or increase the proportion of $D_2R$ homomers. There was no change in BRET50 (an indicator of protomer-protomer affinity) relative to WT for any of the mutants. Treatment of cells with the $D_2R$-selective agonist, quinpirole, did not significantly impact either the BRETmax or BRET50 of WT or mutant $D_2Rs$ (Supplementary Fig. 3). To further understand how these sites impact receptor homomer interactions, computational simulations of V96C (Mut1) and V96S/V97C (Mut2) homodimerization were carried out and predicted that these mutated homodimers exhibit the same architecture as WT, though in Mut2 the interface is tighter than in WT or Mut1 (Supplementary Fig. 4). In the predicted Mut1 dimer, C96 interacts with both Y93 and C96 on the facing protomer, whereas in the Mut2 dimer C97 interacts with C97 and S96 interacts with Y93 on the facing protomer.

### Super-resolution imaging of cell surface $D_2R$ at the single molecule level reveals increased oligomerization in $D_2R$ V96S/V97C mutants

Both BRET and Western blot methods assess the total receptor population within a cell, whether in the biosynthetic, plasma membrane, or endocytic compartments. To quantitate the organization of individual receptors at the plasma membrane, photoactivated localization microscopy with photoactivatable dyes (PD-PALM) with TIRF-imaging was employed. This single-molecule high-resolution technique gives <10 nm resolution and has provided detailed information on plasma membrane localized GPCRs in monomeric, dimeric, or oligomeric forms[36–40]. PD-PALM imaging was carried out on HEK293 cells expressing WT or mutant $D_2Rs$ and directly labeled with an anti-FLAG antibody conjugated to CAGE 500 at a 1:1 ratio. Precise localization of individual receptors and subsequent processing through near neighborhood

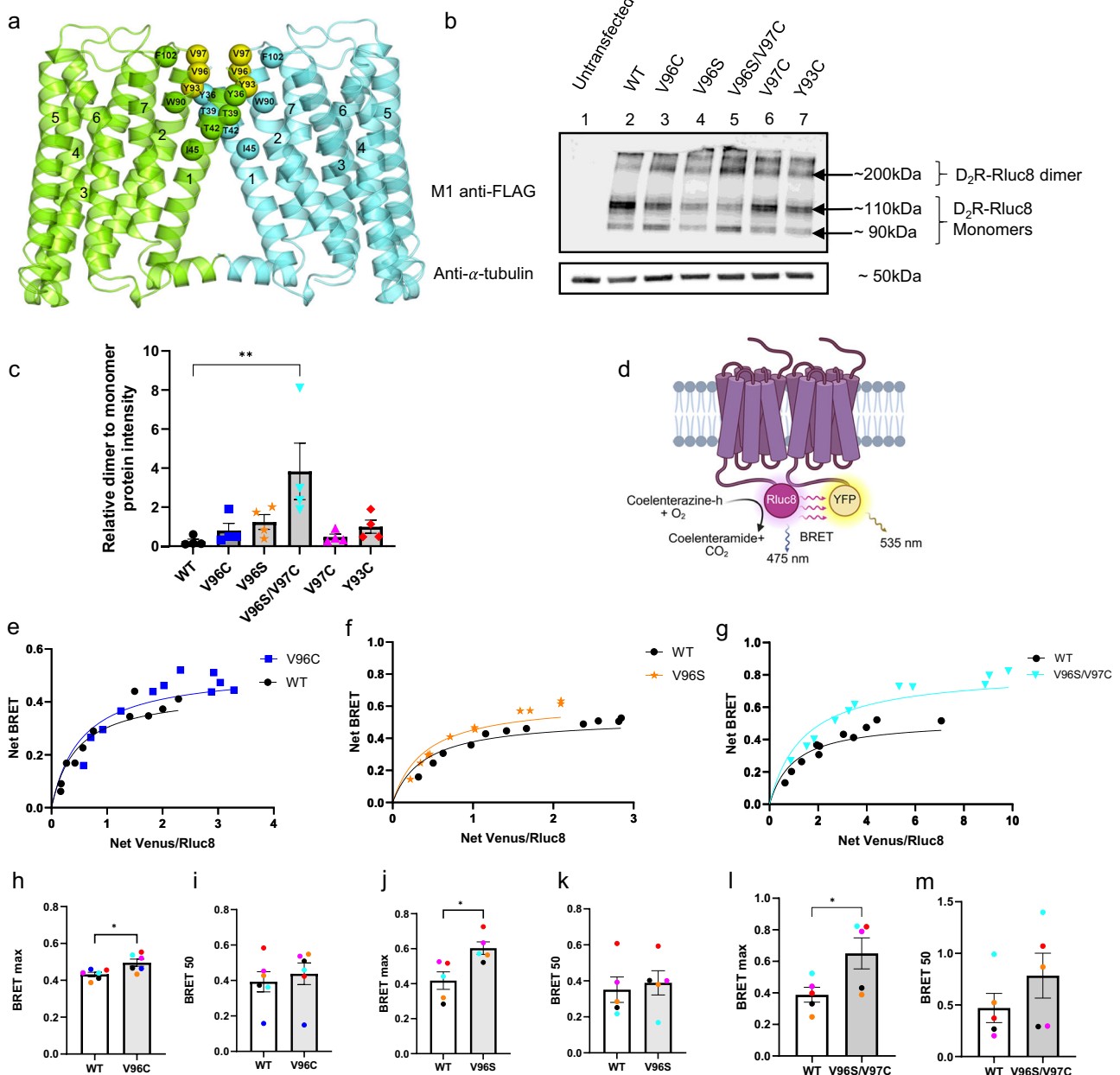

**Fig. 1 | The impact of D$_2$R mutants identified from molecular docking simulations on homodimer stability. a** The predicted WT D$_2$R homodimer (H1, H2 dimer) is shown in cartoon representation, with the two protomers colored lemon-green and aquamarine. Interface amino acids are shown as spheres centred on the Cα-atoms. The mutated amino acids are highlighted in yellow. **b** Western blot analysis of HEK293 cell lysates transfected with WT or mutant D$_2$R-Rluc8 as stated, carried out under reducing conditions. Probed with M1 anti-FLAG to detect the FLAG tag at the N-terminus of receptor constructs (D$_2$R-Rluc8 monomer ~90 kDa or ~110 kDa with glycosylation) and anti-α-tubulin as a loading control. **c** Quantification of band intensity, represented as relative fold change of D$_2$R dimer levels (~200 kDa band) to monomer levels (~110 kDa) under reducing conditions, $N = 4$ independent experiments, error bars are ± SEM. Ordinary one-way ANOVA used to measure statistical differences, ($p** = 0.0051$) **d** Schematic showing BRET assay. Created in BioRender. Sharrocks, K. (2025) https://BioRender.com/u51csyz **e** Representative BRET saturation curve obtained from HEK293 cells expressing either D$_2$R WT (black), D$_2$R V96C (blue), **f** V96S (orange), or **g** V96S/V97C (cyan). Saturation curves were used to quantify BRETmax (**h, j, l**) and BRET50 values (**i, k, m**), color-coded for each individual experiment. Mean ± SEM. $N = 5$ for WT vs V96S and V96S/V97C or $N = 6$ for WT vs V96C. Unpaired, two-tailed Student's t test used to measure statistical differences in BRET50 or BRETmax values between WT and mutant D$_2$R, ($p* = 0.0186$ (**h**) or $0.0162$ (**j**) or $0.0432$ (**l**)).

analysis enabled individual monomer or oligomer populations to be quantified (Fig. 2a, b). Under basal conditions, the percentage of total receptors in a monomeric state in D$_2$R V96S/V97C cells was significantly lower than WT D$_2$R (Fig. 2c). Additionally, for V96S/V97C there was a corresponding significant increase in the proportion of receptors in the di/oligomeric state compared to WT D$_2$R. When these di/oligomer populations were assessed according to the number of receptors in the complex, this revealed that these assemblies primarily involved receptor

dimers (~17% of total receptors), yet also a range of higher-order receptor oligomers. The increase in associated V96S/V97C receptor was not attributed to enrichment of a specific oligomer population but rather across receptor di/oligomers (Fig. 2d).

Following quinpirole stimulation, the different receptors exhibited distinct levels of monomer and oligomeric complexes, compared to basal conditions (Fig. 2e–g). Stimulation of D$_2$R WT expressing cells with quinpirole did not significantly alter the proportion of receptors in an

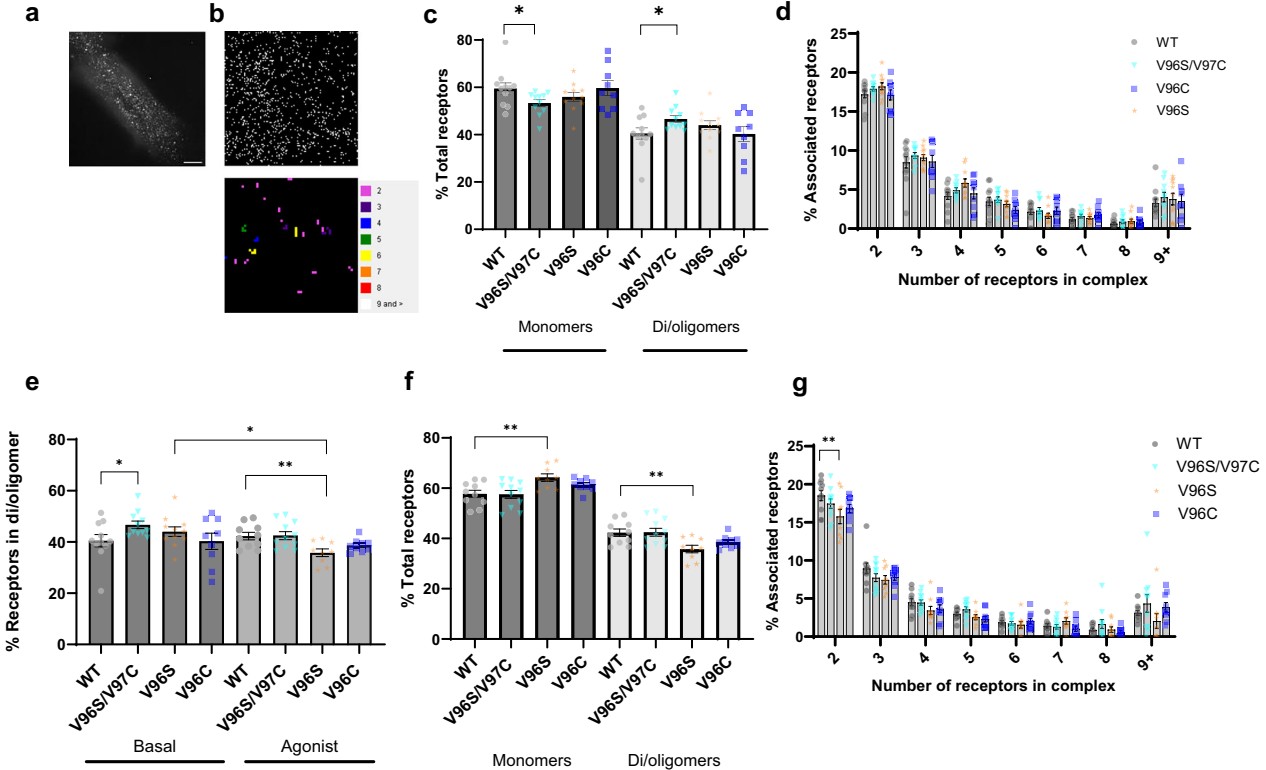

**Fig. 2 | PD-PALM super-resolution microscopy allows quantification of cell surface D$_2$R oligomers. a** Representative image of HEK293 cells transiently transfected with FLAG-D$_2$R and labeled with FLAG-CAGE500, imaged over 20,000 frames by TIRF microscopy following uncaging and activation of the CAGE500 dye. Scale bar = 50 μm. **b** Representative PD-PALM 7 μm$^2$ image of FLAG-D$_2$R labeled with FLAG-CAGE500 with a heat map quantifying oligomer populations. **c** Proportion of D$_2$R WT or mutant receptors as monomers or in di/oligomeric complexes as a percentage of total receptors. **d** Composition of D$_2$R oligomers separated into number of receptors in the complex and expressed as a percentage of total receptors. **e** Proportion of associated receptors in di/oligomeric complexes, expressed as a percentage of total number of receptors under basal conditions or after 5 min 10 μM quinpirole stimulation. **f** Proportion of D$_2$R WT or mutant receptors as monomers or in di/oligomeric complexes as a percentage of total receptors after 5 min 10 μM quinpirole stimulation. **g** Composition of D$_2$R oligomers after 5 min 10 μM quinpirole stimulation prior to fixation, separated into number of receptors in the complex and expressed as a percentage of total receptors. For (**c**–**g**), mean ± SEM, $N = 3$ individual experiments, 3–4 cells imaged for each independent repeat. In **c**, **e**, **f** unpaired, two-tailed Student's $t$ test used to measure differences between WT and mutant D$_2$R ($p^* = 0.042$ (**c**, **e**) or 0.0035 $p^{**} = 0.0068$ (**e**, **f**)). In **d**, **g** two-way ANOVA followed by Šídák's multiple comparisons test used to measure statistical differences between WT and mutant D$_2$R in different receptor complexes ($p^{**} = 0.0019$).

oligomeric complex. However, ligand activation of D$_2$R V96S resulted in a significant decrease in the proportion of receptor in an oligomer compared to both WT D$_2$R expressing cells, whose complexes were not significantly altered when stimulated with ligand (Fig. 2e, f). For V96S, this agonist-dependent decrease in associated receptors was due to a decrease in receptor dimers (Fig. 2g). While PD-PALM analysis suggests that quinpirole treatment may alter di/oligomerization of certain D$_2$R mutants, quinpirole stimulation did not alter BRET saturation profiles (Supplementary Figs. 3 and 5). This could reflect that PD-PALM imaging quantifies complexes of cell surface receptors, while BRET detects interactions across all subcellular compartments.

We have previously demonstrated that receptor density can influence oligomerization for Class A GPCRs as measured by PD-PALM, particularly higher order complexes (>5 receptors/oligomer)[40]. For D$_2$R, the percentage of monomeric receptors was inversely proportional to the receptor density (Supplementary Fig. 5a, b). While for some GPCRs, receptor density impacts higher order complexes[39], our observation for D$_2$Rs via super-resolution imaging is in line with other studies with D$_2$R$_L$ using diffraction-limited single molecule imaging, reporting a small but significant impact of receptor density on the levels of monomer and di/oligomer distribution[14]. Therefore, based on the correlation observed via PD-PALM analysis, the di/oligomeric state of receptors were also analysed under "high" (>100 receptors/μm$^2$) or "low" (<100 receptors/μm$^2$) levels of expressed surface receptors, to understand the effect of

receptor density on homomer formations in WT and mutant receptors (Supplementary Fig. 5b–h). In the absence of agonist, differences in complex formation between WT and mutant receptors were independent of receptor density. The range of receptor densities in cells treated with quinpirole was generally lower for cells expressing D$_2$R V96S and V96C compared to WT or V96S/V97C, likely due to ligand-induced receptor internalization (Supplementary Fig. 5b). This may partially account for the decrease in oligomeric receptors in D$_2$R V96S or V96C compared to WT D$_2$R as when cells with similar receptor densities were analysed, this decrease was not significant (Supplementary Fig. 5g, h).

Taken together, this may suggest that the V96S/V97C mutant increases the propensity for D$_2$R to form an oligomer. This is in line with docking simulations that predict a tighter interface in the V96S/V97C dimer compared to the WT or the V96C dimer (Supplementary Fig. 4). These results provide further evidence that the V96S/V97C mutant D$_2$R may increase the stability of D$_2$R oligomers. While quinpirole stimulation does not significantly alter the organization of D$_2$R complexes.

## D$_2$R mutants that stabilize homomeric interactions impact Gαi-mediated signaling and βarr2 recruitment profiles

Following the observation that some mutations increase the stability and/or degree of D$_2$R homomers, the impact of these mutations on canonical Gαi signaling was assessed. Forskolin-induced cAMP levels were measured in the presence of quinpirole to assess inhibition of

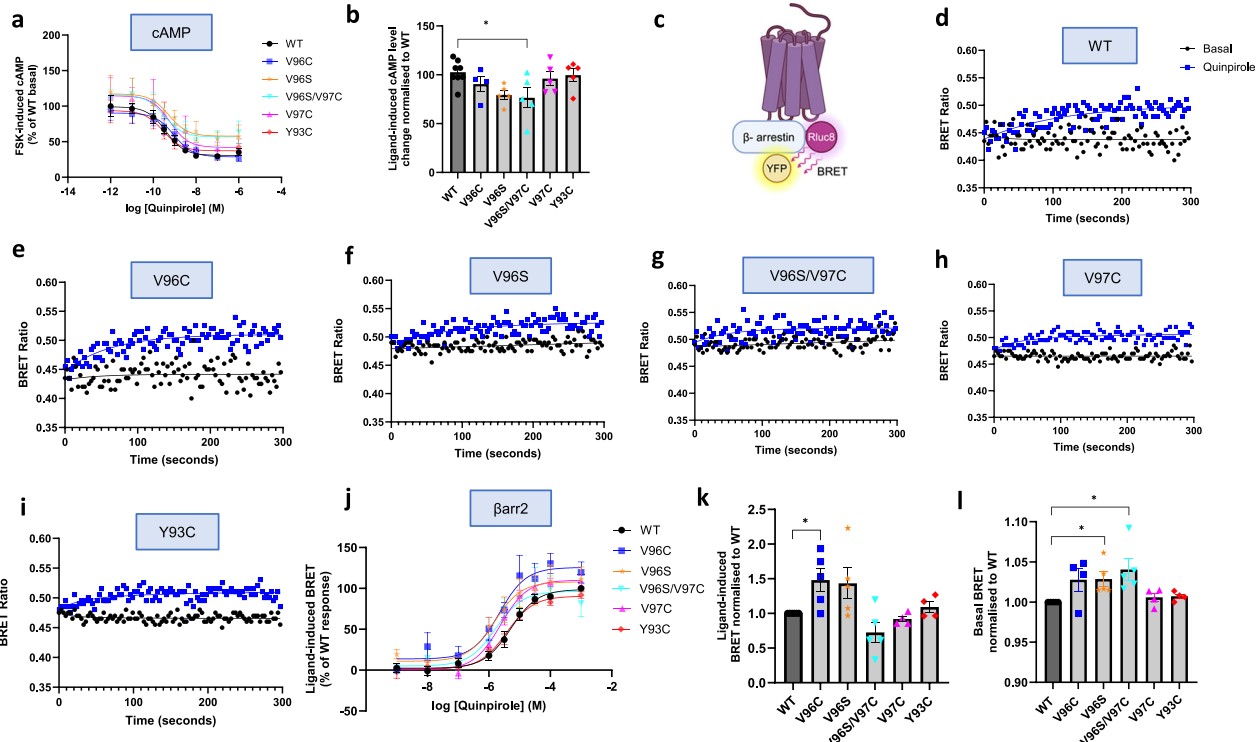

**Fig. 3 | Altered Gαi-signaling and βarr2 recruitment profiles of wildtype and mutant D₂R. a** Forskolin-induced cAMP levels as a percentage of basal cAMP levels of corresponding D₂R WT, quinpirole concentration response curves in D₂R WT (black) or mutant D₂R transfected HEK293 cells. **b** Plot of quinpirole-induced cAMP levels, calculated as a change in cAMP levels in from cells treated with the maximum to the minimum dose of quinpirole, represented as a percentage of WT. In **a**, **b**, values plotted are ± SEM of four (V96C) or five independent experiments for all other D₂R mutants and 8 for WT. One-way ANOVA and Dunnett's multiple comparisons test used to measure statistical differences between WT and mutant maximal responses ($p* = 0.00282$ (V96S/V97C)). **c** Schematic showing BRET assay. Created in BioRender. Sharrocks, K. (2025) https://BioRender.com/u51csyz. Representative kinetic curve showing BRET ratios in HEK293 cells expressing βarr2-YFP and D₂R WT (**d**), V96C (**e**), V96S (**f**), V96S/V97C (**g**), V97C (**h**), Y93C (**i**). Plotted over time for basal (black) and after 10 μM quinpirole stimulation (blue). Graphs representative of 4–6 independent experiments. **j** BRET luminescence values were measured before agonist addition and after addition of 1 nM–1 mM quinpirole, values are average of measurements taken over 5 min. Represented as a percentage of D₂R WT maximal response, for each individual experiment. Mean ± SEM. $N = 12$ for WT, 5 for V96S and V96S/V97C and 4 for Y93C and V97C. See Source Data File for EC₅₀ comparisons. **k** To determine the efficacy of quinpirole to induce βarr2 recruitment, for each individual experiment, ligand-induced BRET values were plotted for cells treated with the maximum dose of quinpirole (1 mM), represented as a fold change from D₂R WT. Basal BRET (**l**) of cells with no ligand and only PBS added prior to luminescence measurement for each individual experiment was plotted and presented as a fold change from WT. One sample t-test and Wilcoxon test. $N = 4$ (V97C and Y93C) or 5 (V96C, V96S and V96S/V97C), ±SEM. ($p* = 0.0488$ (**k**) or 0.0344 or 0.0398(**l**)).

intracellular cAMP levels via D₂R-mediated Gαi signaling. HEK293 cells expressing D₂R WT and stimulated with 1 μM quinpirole for 5, 15, or 30 min exhibited a decrease in forskolin-induced cAMP levels at all agonist time points (Supplementary Fig. 6). This decrease in cAMP is specifically due to D₂R-mediated signaling, as untransfected HEK293 cells do not exhibit this decrease upon quinpirole addition. Concentration-response curves in cells expressing WT or mutant D₂R showed similar curves and EC₅₀ values, suggesting that the potency of quinpirole to activate D₂R/Gαi-mediated signaling pathways is not altered by these mutations (Fig. 3a and Table 1). However, when the ligand-induced Gαi response was calculated from the change in cAMP levels in cells treated with the minimum to the maximum dose of quinpirole, the D₂R V96S/V97C mutant had significantly reduced ligand-induced Gαi response compared to WT (Fig. 3b), suggesting a decrease in efficacy for quinpirole in D₂R V96S/V97C expressing cells.

To assess the impact of these mutant D₂Rs on βarr1/2 recruitment, BRET assays were carried out in HEK293 cells expressing D₂R and βarr1/2. The kinetics of βarr1/2 recruitment were analysed in real time via BRET, measured over a time-course of 5 min following agonist stimulation. The WT D₂R did not recruit βarr1 following quinpirole treatment (Supplementary Fig. 7a–c). However, in cells expressing D₂R WT and βarr2-YFP, there was a rapid increase in BRET levels specifically after

quinpirole addition, up to a maximum level of BRET signal which remained stable for at least 5 min (Fig. 3d and Supplementary Fig. 8). All D₂R mutants (except for D₂R-Rluc8 V96S/V97C) showed a similar profile (Fig. 3e–i). The ability of D₂R to couple to βarr2 over βarr1 is consistent with other studies[41,42]. Calculation of ligand-induced BRET values and subsequent curve-fitting of the BRET profile allowed quantification of the kinetics of βarr2 recruitment to WT and mutant D₂Rs (Table 2 and Supplementary Fig. 8). There was no significant difference in the plateau of the fitted kinetic profiles between WT or any mutant D₂R. However, the halftime of the D₂R V96C kinetic curve was significantly lower than WT, suggesting that D₂R V96C exhibits altered βarr2 recruitment.

To assess if the potency of quinpirole to induce βarr2 recruitment to mutant receptors was altered, cells were treated with increasing concentrations of quinpirole (Fig. 3j–l). D₂R V96C and V96S receptors, but not V97C or Y93C, exhibited a leftward shift in the concentration-response curves, compared to WT with significantly lower EC₅₀ values (Table 1). This suggests that in cells expressing the V96C and V96S mutants, quinpirole exhibits increased potency towards βarr2 recruitment. Additionally, there is a significant increase in basal βarr2 recruitment in cells transfected with D₂R V96S and V96S/V97C compared to D₂R WT (Fig. 3l). Combined with the reduced ligand-induced BRET signal in kinetic assays (Fig. 3), this may suggest increased

**Table 1 | Quinpirole potency in cAMP and βarr2 BRET recruitment assays shown as −logEC$_{50}$**

| Receptor | cAMP −−log EC$_{50}$ | βarr2 recruitment −log EC$_{50}$ |
|---|---|---|
| D$_2$R WT | −9.418 (CI −9.761 to −9.073) | −5.362 (CI −5.527 to −5.196) |
| D$_2$R V96C | −9.036 (CI −9.535 to −8.546) | −5.695 (CI −5.814 to −5.576)* |
| D$_2$R V96S | −9.275 (CI −10.280 to −8.358) | −5.804 (CI −6.044 to −5.565)* |
| D$_2$R V96S/V97C | −9.490 (−10.270 to −8.687) | −5.622 (−5.788 to −5.457) |
| D$_2$R V97C | −9.225 (CI −10.020 to −8.357) | 5.626 (CI −5.167 to −6.0850) |
| D$_2$R Y93C | −9.547 (−10.180 to −8.920) | 5.435 (CI 5.242 to 5.628) |

Data shown with 95% confidence interval (CI). Two-tailed, unpaired Student's $t$-test used to determine statistical significance between WT and mutant −logEC$_{50}$ values ($p* = 0.0147$ (V96C) or 0.0433 (V96S)).

**Table 2 | Quinpirole- induced βarr2 BRET recruitment kinetics**

| | k (second-1) | Initial rate (BRET ratio units per second) | Half time (seconds) | Plateau |
|---|---|---|---|---|
| **WT** | $1.076 \times 10^{-2}$ ($\pm 1.283 \times 10^{-3}$) | $5.27 \times 10^{-4}$ ($\pm 1.644 \times 10^{-4}$) | 68.38 ($\pm 8.537$) | 0.04908 ($\pm 1.166 \times 10^{-2}$) |
| **V96C** | $1.999 \times 10^{-2}$ ($\pm 1.446 \times 10^{-3}$)** | $1.002 \times 10^{-3}$ ($\pm 1.842 \times 10^{-4}$) | 35.55 ($\pm 3.058$)** | 0.05012 ($\pm 8.027 \times 10^{-3}$) |
| **WT** | $4.754 \times 10^{-3}$ ($\pm 3.549 \times 10^{-3}$) | $1.579 \times 10^{-4}$ ($\pm 1.049 \times 10^{-4}$) | 30.28 ($\pm 4.99$) | 0.03682 ($\pm 7.335 \times 10^{-3}$) |
| **Y93C** | $3.33 \times 10^{-3}$ ($\pm 1.943 \times 10^{-3}$) | $1.135 \times 10^{-4}$ ($\pm 5.866 \times 10^{-5}$) | 34.57 ($\pm 14.05$) | 0.03119 ($\pm 7.849 \times 10^{-3}$) |
| **V97C** | $4.434 \times 10^{-3}$ ($\pm 3.577 \times 10^{-3}$) | $1.807 \times 10^{-4}$ ($\pm 1.499 \times 10^{-4}$) | 32.66 ($\pm 4.764$) | 0.0297 ($\pm 6.017 \times 10^{-3}$) |
| **WT** | $3.065 \times 10^{-2}$ ($\pm 1.072 \times 10^{-2}$) | $8.603 \times 10^{-4}$ ($\pm 5.241 \times 10^{-4}$) | 41.08 ($\pm 7.252$) | 0.02122 ($\pm 5.301 \times 10^{-3}$) |
| **V96S** | $2.702 \times 10^{-2}$ ($\pm 8.711 \times 10^{-3}$) | $7.002 \times 10^{-4}$ ($\pm 3.117 \times 10^{-4}$) | 39.36 ($\pm 8.69$) | 0.02394 ($\pm 3.826 \times 10^{-3}$) |

Rate ($k$), initial rate, halftime and plateau were calculated following curve fitting from the waveform produced from ligand–induced BRET recruitment assays (Supplementary Fig. 6). Data are presented with 3 different D$_2$R WT datasets, to allow comparisons with mutant D$_2$R experiments that were carried out at the same time under the same experimental conditions. $N = 4$–6, data shown as ± SEM. Two-tailed, unpaired Student's $t$-test used to determine statistical significance between WT and mutant −logIC$_{50}$ values ($p** = 0.0014$ or 0.0068).

constitutive association between the V96S/V97C receptor and βarr2. These signaling changes were not due to alterations in cell surface receptor expression, as reducing D$_2$R WT cell surface expression to levels obtained with the mutant receptors had no impact on the signal activity profile (Fig. 4 and Supplementary Fig. 9). Given that certain D$_2$R mutants exhibit both stabilized/enhanced homomerization and a potential bias to βarr2, we then assessed whether UNC9994, a D$_2$R-selective ligand, which has been reported to show biased βarr2 signaling in some cell contexts, impacted D$_2$R WT signaling and homomerization (Supplementary Fig. 10)[43,44]. A significant decrease in D$_2$R-mediated Gαi signaling was observed, and while a more transient kinetic profile was observed for βarr2 recruitment, this was not enhanced compared to quinpirole, suggesting UNC9994 acts as a partial biased agonist in this context. Additionally, UNC9994 did not impact D$_2$R homomer associations as indicated by the BRETmax or BRET50 values, similar to what was observed following quinpirole stimulation (Supplementary Fig. 10e–g).

Overall, this data suggests that V96C, and to some extent V96S, show altered quinpirole-induced βarr2 recruitment to D$_2$R, while V96S/V97C exhibits enhanced constitutive association of βarr2. Furthermore, these receptors exhibit bias towards βarr2 recruitment over Gαi signaling pathways.

### Constitutive and ligand-induced receptor internalization is enhanced in D$_2$R V96S/V97C mutants

β-arrestins play key roles in regulating GPCR signal activity[45,46]. One function is driving receptor-mediated internalization. Given the faster (V96C) βarr2 recruitment and, in particular, the ligand-independent (V96S/V97C) recruitment of βarr2, the cell surface expression of these D$_2$Rs was assessed via flow cytometry and confocal microscopy. While all receptors were shown to traffic to the membrane (Supplementary Fig. 11), the cell surface expression of V96S/V97C was lower than WT D$_2$R (Fig. 4a). Due to the observation that V96S/V97C also shows increased basal βarr2 recruitment, the possibility that this decrease in cell surface expression was due to increased internalization was investigated. The percentage of constitutively internalized receptor was calculated as a percentage

decrease of cell surface expression in cells incubated at 37 °C following live antibody labeling, compared to cells incubated at 4 °C to prevent receptor internalization (Fig. 4b). Cells were gated to quantitate levels of D$_2$R in βarr2-YFP-expressing cells. An increase in constitutive internalization of V96S/V97C compared to WT receptor was observed, while Y93C had decreased internalization, and all other mutants showed comparable levels of internalization to cells expressing WT receptor. This increase in constitutive internalization may at least partially account for the decrease in cell surface expression for D$_2$R V96S/V97C compared to WT. There was a significant increase in ligand-induced internalization for all mutant receptors, compared to WT, and this increase was most dramatic in D$_2$R V96S/V97C-expressing cells (Fig. 4c).

The altered ligand-induced and constitutive internalization in mutant D$_2$Rs observed quantitatively by flow cytometry was also confirmed by confocal microscopy (Fig. 4d). Given the decrease in cell surface expression of some D$_2$R mutants, the impact of lower cell surface D$_2$R expression on internalization and signaling was assessed. Transfection of lower amounts of WT D$_2$R plasmid resulted in lower cell-surface receptor expression; however, this decrease in WT expression did not cause a significant change in ligand-induced or constitutive internalization or the efficacy of quinpirole-mediated Gαi signaling (Supplementary Fig. 9c-e), suggesting increased internalization and decreased Gαi signaling in receptor mutants is not solely due to lower cell surface expression. Additionally, reduction of WT D$_2$R expression to a comparable level to the V96S/V97C receptor results in the opposite effect of a decrease in basal associations with βarr2, suggesting that the observed increase in basal BRET associations with βarr2 in V96S/V97C-expressing cells compared to WT is not solely due to lower expression levels of mutant receptor (Supplementary Fig. 9a, b and Fig. 3). This data suggests that enhanced basal βarr2 association may promote both ligand-independent and ligand-induced internalization.

### D$_2$Rs that promote βarr2 recruitment favor βarr2-mediated ERK signaling

Arrestins play well-documented roles in mediating activation of extracellular signal-regulated protein kinase (ERK) signaling[5],

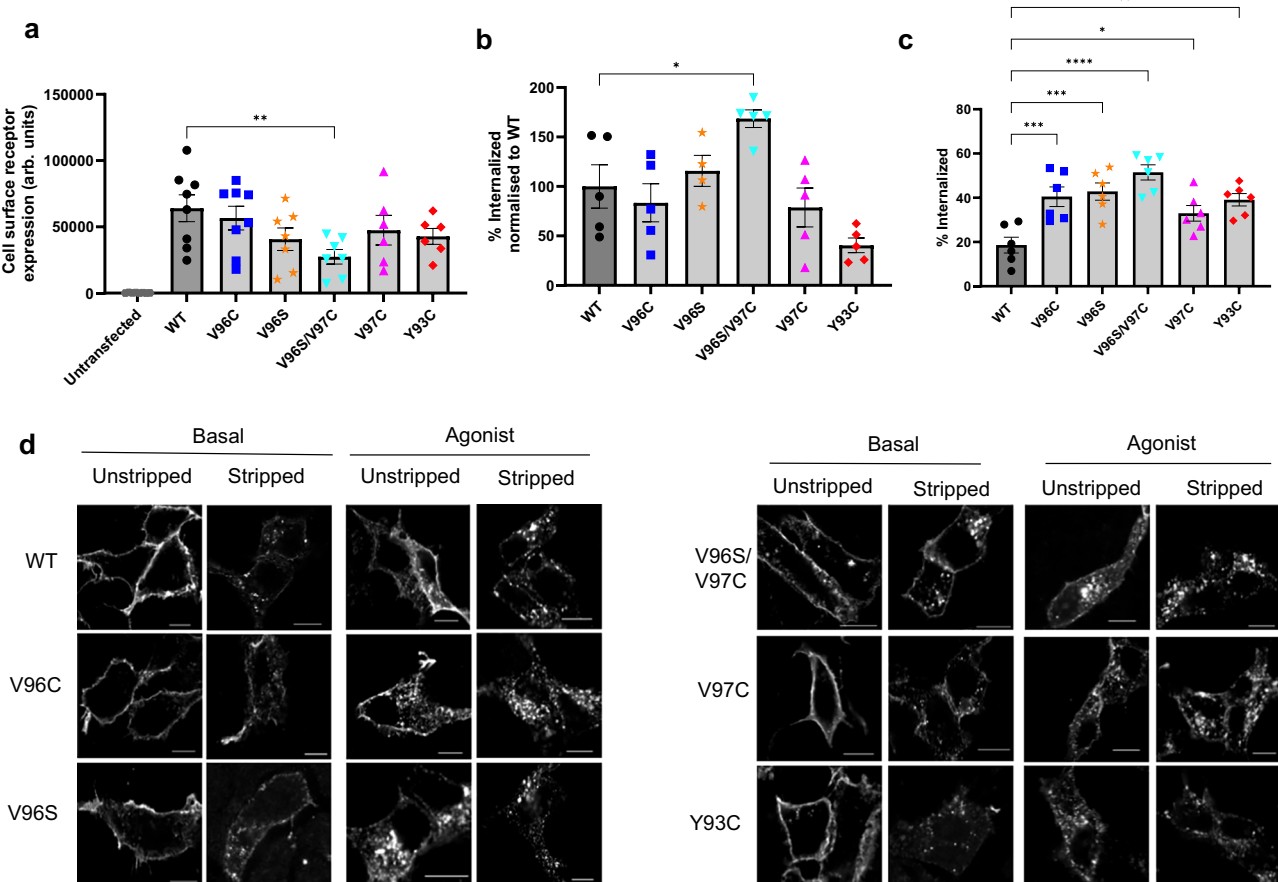

**Fig. 4 | Receptor cell surface expression and internalization assessed by flow cytometry and immunofluorescence microscopy.** HEK293 cells transfected with $D_2R$ constructs were labeled with M1 anti-FLAG primary and Alexa647 conjugated secondary antibody to quantify cell surface receptor expression. **a** Mean fluorescence and % of gated cells were multiplied to give a total value for cell surface $D_2R$ expression. $N = 8$ (WT and V96C), $N = 7$ (V96S and V96S/V97C) or $N = 7$ (V97C and Y93C). **b** In cells expressing βarr2-YFP, constitutive internalization of receptor constructs was calculated as a percentage change of cell surface receptor expression in cells incubated at 4 °C to prevent receptor internalization, compared to cells incubated at 37 °C following antibody labeling, which can continue to constitutively internalize receptor. Expressed as a percentage of WT for each individual experiment. $N = 4$ (V96S) or 5. **c** Quinpirole-induced receptor internalization. Measured as a percentage decrease in cell surface expression level following 10 µM quinpirole stimulation, compared to untreated cells. $N = 6$. In **a–c**, error bars are ± SEM and N represents individual biological replicates. One-way ANOVA followed by Dunnett's multiple comparison test used to measure statistical differences between WT and mutant $D_2R$ ($p^* = 0.0273$ (**b**) or 0.0369 (**c**), $p^{**} = 0.0065$ (**a**) or 0.0018 (**c**), $p^{***} < 0.0009$ or 0.0003, $p^{****} < 0.0001$. **d** Representative confocal microscopy images showing $D_2R$ localization under basal conditions and after 30 min 10 µM quinpirole stimulation, with and without "stripping" of cell surface receptor bound anti-FLAG antibody. Representative of three independent experiments. Scale bar = 10 µm.

therefore, the impact of these mutations on the phosphorylation levels of ERK1/2 was assessed. $D_2Rs$ WT, V96S and V96S/V97C increased ERK activation up to a maximal level following 2–5 min stimulation with quinpirole (Fig. 5a, b). However, V96C exhibited enhanced agonist-dependent ERK signaling compared to WT at the 5-min quinpirole stimulation timepoint, while V96S/V97C exhibited a more rapid ERK signaling profile (Fig. 5a, b). In CRISPR βarr1/2 knockout-cells, the ability of WT $D_2R$ to increase phospho-ERK levels after quinpirole stimulation was maintained (Fig. 5c–e). $D_2R$ V96C, V96S, and V96S/V97C, however, exhibited a decrease in agonist-dependent ERK signaling (Fig. 5f–h and Supplementary Fig. 12). This sensitivity/requirement of βarr for ERK signaling could be due to the altered βarr2 association profiles of these $D_2R$ mutants, which also in turn exhibit enhanced/stabilized homomer associations.

**Differential stoichiometry of $D_2R$ dimers with its G proteins or arrestin**

Insights into the architecture of the assemblies between dimeric $D_2R$ and heterotrimeric Gαi or βarr2 were inferred from molecular modeling. By fitting the $D_2R$ from the cryoEM structure in complex with heterotrimeric Gαi (PDB: 8IRS) onto both protomers in each of the predicted $D_2R$ homodimers, a 2:1 $D_2R$:Gαi stoichiometry is predicted (Supplementary Fig. 13). Steric clashes occur between two G-protein heterotrimers simultaneously docked to the $D_2R$ homodimer, making a 2:2 stoichiometry unlikely. This suggests that for dimeric $D_2R$ the proper $D_2R$:Gαi stoichiometry is 2:1, and any stabilization at the H1-H2 $D_2R$ homodimer, especially V96S/V97C, is expected to reduce the number of receptors available for binding to heterotrimeric Gαi, thus reducing coupling efficiency. The 2:1 $D_2R$:Gαi stoichiometry is in line with the minimal signaling unit found to maximally be activated by agonist binding to a single protomer in an asymmetrical activated $D_2R$ dimer[47].

Docking simulations between the homodimers of $D_2R$ WT, V96C, and V96S/V97C and two different structural models of βarr2 always predicted receptor-arrestin complexes similar to the cryoEM complexes (see "Methods"). It is worth noting that in all dimeric complexes, the $D_2R$ is in an active state as it was extracted from the cryoEM complex with the agonist ritigotine and heterotrimeric Gi (PDB: 8IRS). Remarkably, the architecture of the H1-H2 $D_2R$ dimers favors the simultaneous recruitment of two βarr2 molecules (i.e., with a $D_2R$:βarr2 2:2 stoichiometry) (Fig. 6a, b). Noticeably, the finger loop holds a different conformation in the two βarr2 models, which was

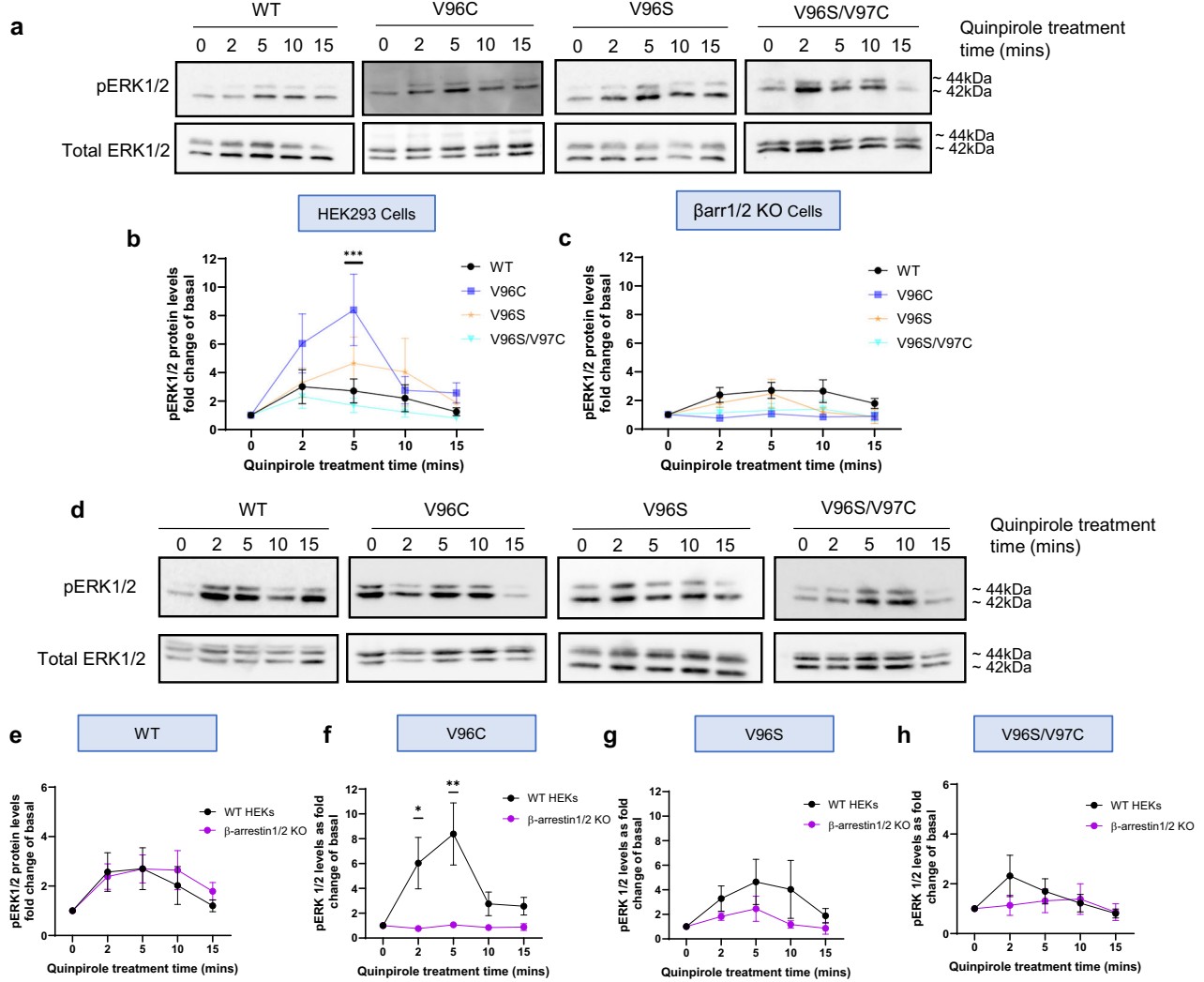

**Fig. 5 | Ligand induced ERK1/2 activation in HEK293 cells and βarr 1/2 knockout cells.** Representative Western blots on HEK293 cells (**a**, **b**) or HEK293 βarr 1/2 KO (βarr 1/2 KO) cell lysates (**c**, **d**) transfected with $D_2R$ constructs and treated with 10 μM quinpirole for 2, 5, 10 or 15 min, probed with phospho-ERK 1/2 and total ERK 1/2 antibodies. Normalized ligand-induced phospho-ERK 1/2 levels expressed as a fold change of basal levels quantified in HEK293 (**b**), or βarr 1/2 KO cell lysates (**c**), transfected with $D_2R$-Rluc8 WT (black) vs V96C (blue), V96S (orange), or V96S/V97C (turquoise) constructs. Direct comparison of relative

pERK1/2 levels in WT HEK293 (black) or βarr 1/2 KO cells (purple) transfected with $D_2R$ WT (**e**), V96C (**f**), V96S (**g**), V96S/V97C (**h**). Mean ± SEM, $N = 5$ (V96S and V96S/V97C) or 6 (V96C) for mutant receptors in HEK293 cells and 4 (V96C and V96S/V97C) or 5 (V96S) in βarr 1/2 KO cells and $N = 8$ for WT $D_2R$. Two-way ANOVA followed by Šídák's multiple comparisons test used to measure statistical differences between WT and mutant $D_2R$ (in **b** p***<0.0005, in **f** $p* = 0.0439$ and p** = 0.015).

inherited from the two different βarr1 structural templates, the finger loop being unstructured in 6TKO while holding one α-helical turn in 7SRS. The 2:2 receptor: βarr2 has been reported in a number of other GPCR dimers, including mGluR3 and mGluR8 and the adhesion GPCR ADGRE1[48–50].

In the predicted complexes between WT or mutant $D_2R$ dimers and both βarr2 structural models, the central-crest loops of active βarr2 form a docking site for IL2 of the receptor, which holds a two-turn α-helix. Specifically, the finger loop of βarr2 docks in the receptor site contributed by the cytosolic extensions of H3 and H6, as well as IL1, IL2, and H8. Furthermore, the middle loop and the β-hairpin that includes the C-loop of βarr2 make contacts with IL2 and H5 of the receptor (Fig. 6a, b). The intermolecular contacts described above define the "core" interactions[51]. While the interface described above is common to all prototypical GPCRs, an additional interface between the N-terminal domain of βarr2 and the phosphorylated tail of the receptor is formed only when such tail is present, which is not the case of $D_2R$.[52] Lack of the phosphorylated C-tail in $D_2R$ may have an effect on

the process of βarr-receptor recognition but not on the final architecture of the complex, which is expected to be the same independent of the presence or absence of the C-tail, i.e., characterized by only core interactions. Therefore, the $D_2R$ dimer-stabilizing mutants both reduce the number of receptors available for Gαi coupling, whilst favoring βarr2 coupling. Measurement of $D_2R$ homomer interactions via BRET saturation assays in βarr1/2 KO cells suggested that the lack of βarr1/2 did not significantly alter $D_2R$ dimerization (Supplementary Fig. 14). This may suggest that $D_2R$ dimers form distinct conformations in the presence or absence of βarrs, without changing overall dimer stability.

The analysis of the available cryoEM structural complexes between GPCRs and βarr1 highlighted their high structural plasticity, which was essentially accounted for by the inclination of βarr with respect to the receptor main axis (Tilt index) and the rotation of βarr parallel to the membrane plane (Rot index)[51]. According to the Tilt and Rot indices, both the predicted complexes between $D_2R$ and βarr2 resemble more the complexes between βarr1 and the β1-adrenergic

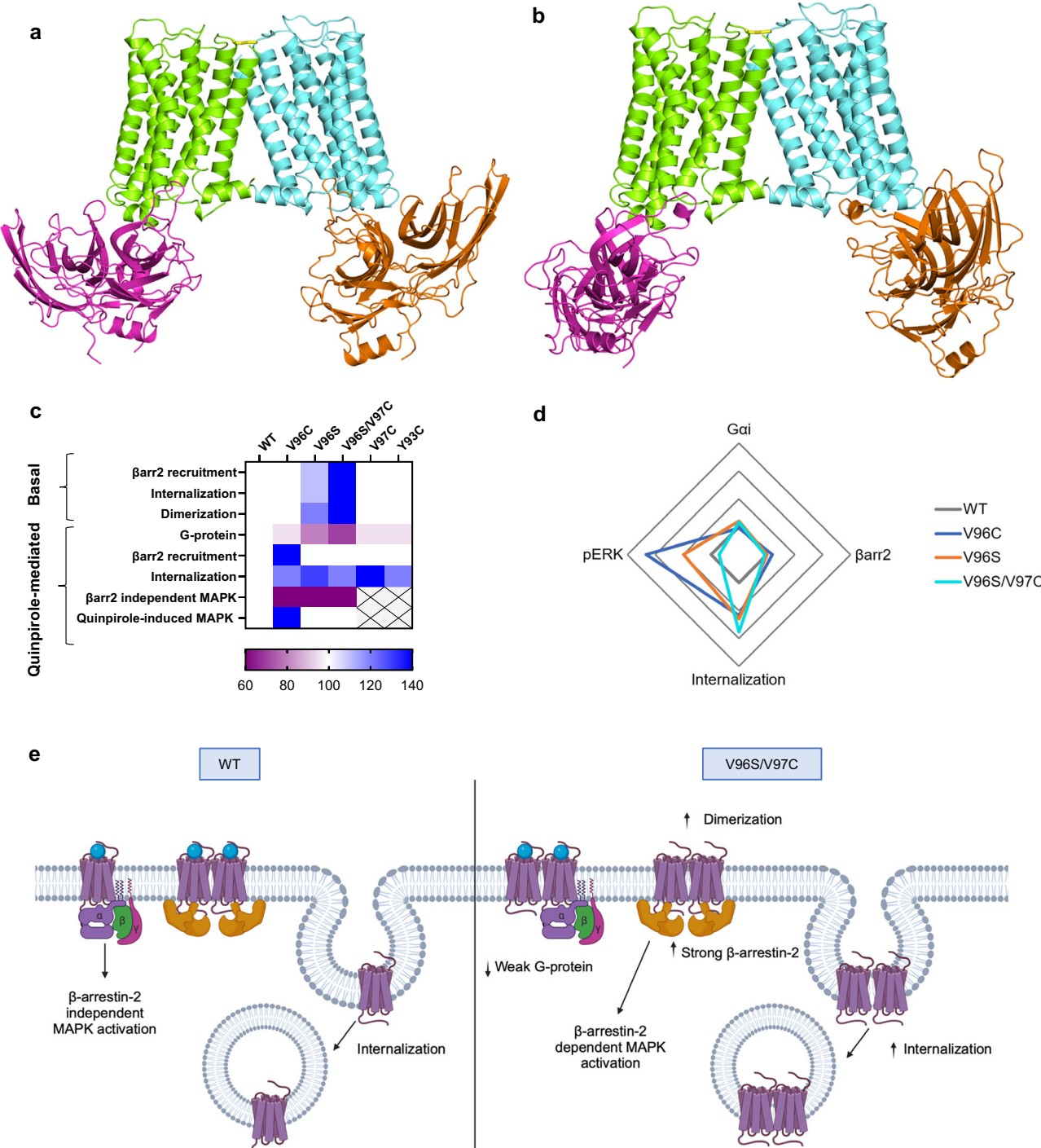

**Fig. 6 | Predicted models for D₂R homodimer-βarr2 complexes and summary schematics.** Cartoon representations of two different structural complexes between the V96S/V97C mutant D₂R homodimer (protomers lemon-green and aquamarine) and βarr2 (orange and magenta) are shown. The structural models of βarr2 shown in the **a**, **b** panels have been, respectively, achieved by using the βarr1 structures from the 6TKO and 7SRS cryoEM complexes as templates. The side chains of C97 are shown in yellow sticks; the distance between the two sulfur atoms is 2.0 Å. **c** Heat map summarizing the functional effects of D₂R mutants relative to WT. 100 is WT and white, an increase relative to WT is blue and a decrease is purple. **d** Quinpirole-induced Gi signaling, βarr2 recruitment, quinpirole-induced internalization, and pERK activation of D₂R mutants as a fold change relative to D₂R WT. G-protein and β-arrestin bias calculated as Emax/EC₅₀ **e** Summary schematic of proposed βarr2 biased signaling from D₂R V96S/V97C oligomers. Created in BioRender. Sharrocks, K. (2025) https://BioRender.com/0uuul4q.

receptor (β₁AR) or the 5HT₂B-serotonin receptor (5HT₂BR) than the complexes with the other GPCRs (i.e., the m2-muscarinic receptor (M₂R, PDB: 6U1N), the NTS1 neurotensin receptor (NTS₁R, PDB: 6PWC), the V2-vasopressin receptor (V₂R, PDB: 7R0C), and the CB1-cannabinoid receptor (CB₁R, PDB: 8WRZ)). By overlapping the M₂R, NTS₁R, V₂R, and CB₁R in complex with βarr1 with both D₂R protomers in the predicted dimer, no clash between the βarr molecules was observed. This suggests that the H1-H2 architecture of the D₂R dimer in complex with βarr in a 2:2 stoichiometry is compatible with all combinations of Tilt and Rot values explored by the cryoEM GPCR-βarr1 complexes. The high plasticity of the GPCR-βarr1 complexes emerges also from a study on the mGlu8 metabotropic receptor (mGluR8),

revealing an additional interaction core-bound mode of βarr1 not yet observed in other cryoEM complexes[49].

## Discussion

Combining molecular docking simulations with biophysical, super-resolution, and functional studies, we propose a model for stabilizing $D_2R$ homomer associations and demonstrate the utility of such an approach to identify a mechanism for not only stabilizing receptor-receptor associations, but also for promoting both βarr2 interaction and βarr2-mediated functions. This reveals a potential role for $D_2R$ di/oligomerization in βarr2-biased signaling that could be targeted for therapeutic benefits.

The predicted interface in symmetric $D_2R$ homodimers with both protomers either inactive or active, characterized by H1-H1, H1-H2, and H2-H2 contacts (i.e., H1-H2 dimer), is shared with other published high-resolution GPCR dimer structures, including rhodopsin, κ-opioid, β1-adrenergic, μ-opioid, and human apelin receptors[53–57]. Previous studies have suggested that H4 participates in $D_2R$ homodimer interface, but a second interface involving H1 and H8 has also been suggested[29,58,59]. We found the involvement of H4 in some $D_2R$ dimers with both protomers inactive or with asymmetric inactive-active dimers, though with worse membrane topology. Additionally, several studies have suggested that IL1, EL2, and EL3 are important in stabilizing GPCR dimer interactions[59–61]. However, this is based on computational modeling and biophysical data, with no current published dimer structure for the $D_2R$ homodimer. It is also possible that multiple dimer interfaces exist, as reported for the β1AR, where both a H1-H2-H8-EL1 and H4-H5-EL2-IL2 interface exist[53]. Given we observed both dimers and higher-order oligomers, we cannot rule out that $D_2R$ interacts through additional interfaces depending on whether it exists in a dimeric or higher-order complex. Mutations at specific sites of the interface predicted herein increased the stability of $D_2R$ homomers, with the V96S/V97C double mutant exhibiting the greatest shift towards $D_2Rs$ in an oligomeric state in the absence of ligand (~6% increase in all oligomer types compared to $D_2R$ WT), indicating an increase in constitutive oligomer stability. Super-resolution, single molecule analysis revealed there was little change in the proportion of WT and V96S/V97C $D_2Rs$ in an oligomeric state after cells were stimulated with quinpirole. Interestingly, the proportion of V96S receptors in an oligomeric complex at the cell surface was reduced when stimulated with quinpirole, which could reflect that di/oligomers of this $D_2R$ are internalized, consistent with the enhanced ligand-induced internalization and oligomerization favoring βarr2 coupling.

Ligand bias, whereby ligands can promote one pathway over another, is of interest both therapeutically and for understanding the molecular basis of pathway preferences for different GPCRs, in different conformational states[62,63]. In this study, receptor mutants, which showed increased di/oligomerization also showed increased ligand-induced (V96C and V96S) or constitutive (V96S/V97C) associations with βarr2 (Summarized in Fig. 6c–e). It is important to note that while the BRET assay results presented here suggest a steady state recruitment of βarr2, it cannot be ruled out that this represents multiple GPCR-βarr associations and dissociations, perhaps with distinct $D_2R$-βarr conformations[64]. It is also worth acknowledging that it is possible that the mutations may influence the conformations of monomeric receptors and/or protomers within an oligomeric complex, which may partially account for some of the observed functional effects with βarr2 for these $D_2R$ mutants.

There are many factors that can impact the signaling pathway preference of a GPCR, including membrane lipid composition and localization of receptors[36,65–68]. Interestingly, these factors have also been shown to influence oligomerization of GPCRs[69]. While changes in the organization of $D_2R$ complexes at the cell surface detected in PD-PALM experiments may be small, even slight changes in complex formation can have dramatic effects on receptor signaling in cells[38,39]. By

enhancing specific receptor conformations, such as a propensity to oligomerize and associate with βarr2, the role of such conformations on receptor activity may be amplified. These changes are in line with prior studies that have employed PD-PALM to assess GPCR di/oligomerization, where even small changes in protomer composition within a single oligomeric complex can direct how these receptors modulate signaling[38,39].

There is some debate about the role of βarrs in $D_2R$-mediated ERK activation. It is likely that mechanisms are both cell-type specific and $D_2R$-isoform specific, with $D_2R$ long and short differing in the way they mediate MAPK activation[70,71]. Interestingly, the apparent bias towards βarr2 recruitment in $D_2R$ mutants results in enhanced internalization and a βarr2-dependent mechanism for downstream mediation of the MAPK pathway. It is possible that increased dimerization of these mutants makes them more likely to internalize as has been demonstrated from CXCR4 oligomers[19]. Taken together, the most striking changes in signaling under basal conditions are observed with the $D_2R$ V96S/V97C double mutant, which shows a strong preference for βarr2, required for downstream signaling.

In this study, we did not explore the role of GRKs in βarr2 associations. Prior studies have suggested that while phosphorylation of the IL3 of $D_2R$ can enhance βarr2 recruitment, it has also been shown that neither GRK2 kinase activity nor G protein activation may be required for βarr recruitment[72–77]. Thus, it is possible that complexes of $D_2R$ mutants such as V96S/V97C, which exhibit constitutive association with βarr2, may also favor conformations for associations with GRK2 to promote βarr2 interactions and could be explored in future studies. Here we have shown that different $D_2R$ mutants exhibit distinct βarr2 signaling profiles, supporting the broad spectrum of GPCR-βarr conformations that are being uncovered[78].

The impact of oligomerization on biased signaling has focused on heteromers rather than homomers, including the $A_{2A}R$-$D_2R$ heteromer[79,80]. Previous studies have reported that oligomerization can either increase or decrease G-protein activation, suggesting there is no consensus for all GPCRs in one class, but such behaviour is receptor specific[19,21,81,82]. The role of GPCR oligomerization on βarr recruitment has been less well studied. Oligomerization of the PAFR has been shown to cause a bias away from βarr recruitment, while our study demonstrates an opposing role for the $D_2R$[21]. It has been suggested that allosteric interactions between receptor protomers upon receptor activation of the $A_{2A}R$-$D_2R$ heteromer causes the $D_2R$ protomer to favor βarr2 recruitment[80]. It is possible that a similar phenomenon is occurring in the $D_2R$ mutants explored in this study that exhibit increased βarr2 recruitment and associated signaling, and also exhibit enhanced homomer stability.

Dimerization has been suggested to significantly increase the surface area available for arrestins to bind, which may provide a rationale for the increased oligomerization, which we observe in receptor mutants that also show a bias towards arrestin-mediated signaling pathways[11,83]. Consistently, the predicted $D_2R$ dimer is able to simultaneously accommodate two βarr2 molecules. It is possible that the V96S/V97C double mutant and, to some extent, the V96S $D_2R$ mutant, are exhibiting a more 'active-like' conformation in the absence of agonists, which causes them to associate more readily with βarr2, increase internalization, and a dependence upon βarr2 for MAPK signaling. Previous studies have found that βarrs are involved in $D_2R$-mediated ERK phosphorylation in the $A_{2A}R$-$D_2R$ heteromer, but not when $D_2R$ is expressed alone[84]. It is also possible that for these $D_2R$ mutants, increased homo-oligomerization induces increased dependence upon βarr2 for signaling pathways. This is supported by docking simulations between $D_2R$ homodimers and βarr2, which showed a 2:2 receptor: βarr2 stoichiometry while exhibiting a 2:1 receptor: Gi stoichiometry (Fig. 6a, b and Supplementary Fig. 13). This suggests that $D_2R$ dimers, and therefore mutations that stabilize homo-di/oligomers involving H1 at the inter-protomer interface, favor βarr2 coupling over

Gαi coupling. This hypothesis is also supported by the recent cryoEM complexes between dimeric mGluR3 and dimeric apelin receptors and βarr1[50]. These docking simulations may provide an insight into the structural basis behind the observed increase in oligomerization in $D_2R$ mutants that also exhibit increased βarr2 associations. Future studies could explore these complexes further, including capturing high-resolution structures of $D_2R$ dimers in complex with G protein and βarr2 and/or GRKs.

In conclusion, the role of $D_2R$ oligomers in disease makes these complexes attractive therapeutic targets[31,85–87]. The lack of high-resolution structures of the $D_2R$ homodimer has hindered our ability to exploit oligomers for therapeutic benefits. Here, we suggest models for the $D_2R$ homodimer, and experimental validation confirms a key role for H1 and H2 at the predicted $D_2R$ homodimer interface to increase dimer stability, and a bias towards βarr2. These mutant constructs could form the basis for structural studies capturing $D_2R$ homomers and specific $D_2R$- βarr complexes. More broadly, this study provides a framework for engineering the stabilization of Class A GPCR dimers and may suggest that $D_2R$ oligomers favor an "active-like" conformation that promotes βarr recruitment.

## Methods

### Molecular modeling

Prediction of likely architectures of $D_2R$ homodimers relied on a computational approach developed for quaternary structure predictions of transmembrane α-helical proteins, including a number of GPCRs, i.e., FiPD-based approach[88–90]. It consists of rigid-body docking using both a version of the ZDOCK program devoid of desolvation as a component of the docking score (v2.1)[91], followed by employment of a membrane topology filter by the FiPD software to minimize false positives. A rotational sampling interval of 6° was set (i.e., dense sampling), and the best 4000 solutions were retained and ranked according to the ZDOCK score. Such solutions were then filtered according to the "membrane topology" filter, which discards all those solutions that violate the membrane topology requirements. The membrane topology filter, indeed, discards all the solutions characterized by a deviation angle from the original z-axis, i.e., tilt angle, and a displacement of the geometrical center along the z-axis, i.e., z-offset, above defined threshold values, which were 0.4 radians and 6.0 Å, respectively. The filtered solutions underwent cluster analysis by using a 3-Å Cα-atom Root Mean Square Deviation (RMSD) followed by visual inspection of the cluster centers for selecting the final solutions on the basis of membrane topology indices and docking score. Symmetrical $D_2R$ homodimers holding both protomers either inactive or active, and asymmetric inactive-active dimers were predicted. The following structures were probed as sources of $D_2R$ protomers: (a) inactive-state structures: 6CM4, 6LUQ, and 7DFP (i.e., $D_2R$ in complex with risperidone, haloperidol, and spiperone, respectively), and (b) active-state structures: 7JVR and 8IRS (i.e., the ternary complexes between heterotrimeric Gi and $D_2R$ in complex with bromocriptine and ritigotine, respectively). The best results in terms of reliable docking solutions (according to the membrane topology indices) were achieved for the active-state dimers, by using $D_2R$ from 8IRS as a protomer, and for the inactive-state dimers, by using $D_2R$ from 6CM4 completed with the missing IL2 as a protomer. The employment of an active-state structural model of $D_2R$ was also instrumental in predicting supramolecular complexes between $D_2R$ dimers and either βarr1 or βarr2. The latter predictions were carried out by ZDOCK v3.0.2[92] coupled with a FiPD-based filtering approach based on a distance cutoff between the DRY arginine (R132) of $D_2R$ and L/V71 in the finger loops of βarr1 or βarr2, respectively, followed by cluster analysis. In those docking simulations, the receptor was used as a fixed target while βarr1 or βarr2 was used as a probe. The structural model of receptor-bound βarr2 was achieved by comparative modeling (by Modeler[93]), using the cyoEM structures of βarr1 bound either to the $β_1$-AR (PDB: 6TKO) or to the $5HT_{2B}$ receptor (PDB: 7SRS) as templates. Remarkably, the two βarr1 templates differed in the conformation of several loops, including the finger loop, an essential recognition point for receptor binding. Such comparative modeling was necessary because none of the available βarr2 structures, co-crystallized with peptides from the C-tails of a number of GPCRs, represent real receptor-bound forms as far as conformation and arrangements of the loops in the central crest are concerned. Accordingly, the employment of those βarr2 structures in docking simulations with the $D_2R$ did not produce any reliable solution. The employment of two different structural models of βarr2 served to overcome in part the lack of cryoEM complexes of βarr2 with GPCRs and to take into account possible structural variability in the receptor-bound states of the same arrestin. For βarr1, the most complete versions of the cryoEM structures 6TKO and 7SRS were employed. Only the results of βarr2-$D_2R$ docking are shown here. Evaluation of the native-like feature of a docking solution was based on structural comparisons with the cryoEM receptor-βarr1 complexes (6TKO and 7SRS).

### Cell culture and transfection

HEK293 cells (human embryonic kidney cells) (ATCC, CRL-1573) and βarr1/2 knockout cells were maintained as a monolayer in T-75 cm2 flasks (Sigma). βarr1/2 knockout HEK293 cells were made by using CRISPR-Cas9 (kindly provided by Asuka Inoue, Univ. Tohoku). Cells were cultured in Dulbecco's Modified Eagle's Medium (DMEM) media (Gibco) containing 4.5 g/L glucose, 1% (v/v) L-glutamine and supplemented with 10% (v/v) FBS (Gibco), 1% (v/v) streptomycin/penicillin (Sigma) in a humidified atmosphere of 5% $CO_2$ at 37 °C. Cells were usually passaged every 3–4 days in a 1:10 ratio using 0.05% (v/v) trypsin (Gibco, UK) in phosphate buffered saline (PBS). HEK293 cells were seeded and cultured in 6-well plates 1 day before transfection in DMEM with 10% (v/v) FBS (Gibco), 1% (v/v) streptomycin/penicillin (Sigma). Cells were transfected at 80–90% confluency with Lipofectamine 2000 reagent (Invitrogen, UK).

### Plasmid DNA constructs

βarr1-YFP and βarr2-YFP plasmids were obtained from Frederic Jean-Alphonse (CNRS, Nouzilly). FLAG-$D_2R$-Venus and Rluc8 plasmids were obtained from Jonathan Javitch. The FLAG-$D_2R$-Rluc8 plasmid was used to obtain the $D_2R$ Cys mutant constructs using the Agilent Quikchange II site-directed Mutagenesis Kit according to the manufacturer's instructions. Mutant $D_2R$ constructs were introduced into $D_2R$-Venus plasmids via the In-Fusion Snap Assembly cloning kit (Takara) according to manufactures instructions and inserted into FLAG-$D_2R$-Venus DNA using BamHI and Kpn1 restriction sites. See Supplementary table 2 for a list of plasmids used in this study.

### BRET saturation assays

Cells were transfected with constant amounts of $D_2R$-Rluc8 plasmid DNA and increasing amounts of $D_2R$-Venus (0–3 μg in 6 well plates) plasmid and assayed 48 h post-transfection. Cells in suspension were plated in white 96 well plates in triplicate. Coelenterazine-h (Promega) was used at a final concentration of 5 μM. BRET emissions using the short-wavelength filter at 475 nm and long wavelength filter 535 nm were measured in 10 repeat cycles using a LUMIstarOPTIMA plate reader (BMG Labtech). Fluorescence of Venus-tagged receptor was measured in duplicate after excitation at 485 nm and emission at 535 nm.

For each condition, net BRET values were calculated (535 nm value/475 nm value) over 10 cycles, where BRET ratios for the negative control donor only condition were subtracted. Rluc8 values (475 nm emission value) were averaged over the 10 cycles and again for the duplicate repeats. Net Venus values were calculated by subtracting Venus measurements of donor only conditions. Net BRET

values were plotted against Net Venus/Rluc8 using GraphPad Prism 10 and fitted with a non-linear binding curve assuming one site-specific binding.

## β-arr recruitment BRET assays

HEK293 cells were transfected with $D_2R$-Rluc8 plasmids and βarr1/2-YFP constructs. 48 h post-transfection, cells were resuspended in PBS. For BRET time course assays, Venus expression (excitation 485 and emission 535 nm) was measured. For basal BRET emissions, a separate set of duplicate cell suspensions was measured at 475 nm excitation and 535 nm emission wavelengths after addition of 5 μM coelenterazine-h (Promega). PBS or 10 μM quinpirole was added to cell suspensions, and BRET emissions were immediately measured again. Raw BRET ratios were calculated (535 nm value/ 475 nm value). To calculate ligand-induced BRET, basal BRET ratios were subtracted from BRET ratios after agonist addition. For dose response BRET experiments, increasing concentrations of quinpirole agonist were added to sets of duplicate cell suspensions after basal BRET emissions had been measured. After ligand-induced BRET values had been calculated, data were normalized by subtracting the control with only PBS added.

Analysis was based on βarr recruitment waveform methods as previously described[94]. The BRET emissions were measured every 3–5 s for a total of 5 min. Ligand-induced BRET ratios were calculated and plotted against time, and recruitment of βarr was observed to be stable recruitment, where the waveform rises to a steady state. The waveforms were fitted to non-linear one-phase association curve in GraphPad Prism 10 to allow quantification of kinetics. The rate constant s-1 was quantified and can be represented as the halftime, in seconds, (computed as ln(2)/k). The steady-state level of recruitment is represented as the plateau of the waveform, which also indicates the affinity of the receptor for βarr. The initial rate of recruitment was also calculated as k*span (maximum BRET ratio).

## Flow cytometry

Cells were live labeled with M1 anti-FLAG antibody (1:1000 Sigma Aldrich, no.3040) 48 h post transfection for 30 min at 37 °C. For the flow internalization assay, antibody incubation was carried out at 4 °C for 1 h for all cells, then one set of cells were transferred to the 37 °C incubator and allowed to recover for 1 h. Agonist stimulation at 37 °C was carried out 30 min in to the 1-h recovery period. The other set of cells was kept at 4 °C and washed 3 times with cold DMEM media. Cells were resuspended in FACS buffer (PBS with $Ca^{2+}$ and $Mg^{2+}$ and 2% FBS), centrifuged, and the cells resuspended in either FACS buffer alone for untreated cells or with Alexa Fluor 647 antibody (1:2000; Invitrogen, A32728). Cell suspensions were incubated for 1 h on ice and washed in FACS buffer three times. The cell suspensions were transferred to individual round-bottomed polypropylene tubes, and the immuno-fluorescence measured on a FACS Calibur Flow Cytometer (BD Biosciences) and exported into CellQuest Pro.

## cAMP assay

HEK293 cells were transfected with 1 μg of $D_2R$-Rluc8 constructs. After 24 h, approximately 40,000 cells/well were replated into a 96-well format in triplicate for each condition. After a further 24 h, the cells washed with PBS and pre-treated with 0.5 mM IBMX in DMEM with 0.1% Bovine serum albumin (BSA) for 5 min at 37 °C. Forskolin was diluted in 0.5 mM IBMX in DMEM 0.1% BSA to give a final concentration of 3 μM. Cells were stimulated with ligands diluted in the forskolin solution and incubated at 37 °C. After stimulation, cells were washed with ice-cold PBS and lysed. The level of cAMP in lysates was measured using the cAMP Gs Dynamic 2 kit (Revvity).

## Western blotting

Cells were washed with PBS and lysed with ice-cold RIPA lysis buffer (50 mM Tris pH 7.4, 1% Triton X-100, 140 mM NaCl, 5 mM EDTA, 1 mM NaF, 1 mM PMSF, 1 mM sodium orthovandate, and a protease inhibitor tablet (Roche)). Protein concentrations were determined using BSA standard curve and diluted in Laemmli sample buffer (Sigma) in the presence or absence of 5% 2-β-mercaptoethanol. Samples were separated via SDS-polyacrylamide gel and transferred to a nitrocellulose membrane. The membrane was blocked and incubated in primary antibody diluted overnight (See Supplementary Table 1). Membranes were incubated in HRP-linked secondary antibody and imaged using Immobilon Forte Western HRP reagent (Millipore) and X-ray film (Amersham) or Chemi-imager. The band intensities were analysed using ImageJ.

For pERK Western blotting, cells were serum starved for 16 h before quinpirole stimulation and lysis of cells. All lysates were diluted to 20 μg of protein in Laemmli buffer in the presence of 5% 2-β-mercaptoethanol, boiled for 5 min at 95 °C before loading. Quantification of protein intensity was done separately for each blot, and where multiple blots were used the same $D_2R$ WT lysates were employed to allow direct comparison with mutant $D_2R$ on the same membrane. See the Source Data file for examples of uncropped blots.

## Confocal microscopy

Transfected cells were replated onto coverslips in a 24-well format with approximately 75,000 cells/well and left for a further 24–48 h. Cells were incubated in M1 anti-FLAG antibody (Sigma Aldrich, no.3040) diluted 1:500 with or without 10 μM quinpirole for 30 min at 37 °C. To remove the cell surface antibody, cells were washed 3 times with 0.04 M EDTA in PBS-$Ca^{2+}$ before fixation. Cells were then fixed with 4% PFA in PBS-$Ca^{2+}$, permeabilised with 0.2% Triton-X in PBS-$Ca^{2+}$, and blocked for 1 h at RT in 2% FBS in PBS-$Ca^{2+}$ prior to incubation in Alexa Fluor-conjugated secondary antibody (1:000 dilution) (Thermofisher, A32728). Slides were mounted onto coverslips, and imaged were obtained with a Leica Stellaris 8 microscope using a 64x oil immersion objective and sequential excitation at wavelengths of 405, 488, and 647 nm.

## PD-PALM

Transfected cells were plated onto 35 mm dishes (Mattek) with 14 mm × 1.5 mm glass coverslips. Anti-FLAG (Sigma Aldrich) primary antibodies were labeled with photo switchable dye CAGE 500 according to the manufacturer's instructions to give a 1:1 antibody:dye molar ratio (Abberior). Live cells were incubated with FLAG-CAGE500 conjugated antibody, diluted in 10% FCS in PBS $Ca^{2+}$ at 37 °C for 30 min. All steps from antibody addition to imaging were carried out in the dark to prevent activation on the conjugated antibodies. Cells were washed in PBS $Ca^{2+}$ before fixation in 4% PFA and 0.2% Glutaraldehyde for 30 min (Sigma) and maintained in PBS $Ca^{2+}$.

PALM images were acquired using the Zeiss Eyra PS1 microscope with 1.45 numerical aperture, 100x oil immersion objective in total internal reflection fluorescence (TIRF) mode. Photoactivation of CAGE 500 dye was achieved by simultaneous activation with 405 nm (UV light) and excitation and photobleaching with 488 nm laser. Lasers were switched on 30 min before imaging to allow stabilization. Images were captured over 20,000 frames with an exposure time of 100 ms using the ZEN software.

Location analysis of receptors was undertaken using the Quick-PALM plugin in Fiji, as previously described[95]. Two 7 × 7 μm region of interests were analysed within the cell boundary using the QuickPALM plugin. The parameters used were typically a signal-to-noise ratio of 7 and a full-half width maximum of 5, although these were determined for each experiment. This analysis outputs XY coordinates of the localized particles with sub-pixel accuracy. Particles within 20 nm of each other were discounted to prevent overestimation of receptor complexes. The coordinates were inputted into a Java-based PD-interpreter application[39,40]. to determine the number of associated receptor molecules. This app uses Getis-Franklin neighborhood

analysis with a search radius of 50 nm. The software discounts a receptor from further searches once it has been assigned as participating in a homomer, to prevent double-counting. The output is a table containing both the number of associations and the number of receptor molecules in each complex. PD-PALM analysis software is available as Supplementary Data 1.

## Quantification and statistical analysis

All statistical analysis was performed in GraphPad Prism 10. Specific statistical tests are noted in each figure legend. In most cases, two-tailed, unpaired t-tests were used when comparing two groups, or one-way ANOVA with Dunnett's multiple comparison test was used for multiple groups. For comparing two variables and multiple groups, two-way ANOVA followed by Šídák's multiple comparisons test was used. In all tests, $p < 0.05$ was considered statistically significant.

## Reporting summary

Further information on research design is available in the Nature Portfolio Reporting Summary linked to this article.

## Data availability

Source data are provided with this paper. Any additional information relating to this study are available from the corresponding author upon reasonable request. Structures used in this study are available on Protein Data Bank: 6CM4; 6LUQ; 7DFP; 7JVR; 8IRS; 6TKO; 7SRS. Source data are provided with this paper.

## Code availability

Custom codes used to analyse PD-PALM data are provided in Supplementary Data 1.

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

## Acknowledgements

We would like to acknowledge Drs Stephen Rothery and David Gaboriau at the Facility for Imaging of Light Microscopy at Imperial College London for their technical support for microscopy experiments. Dr. Asuka Inoue (Univ. Tohoku, Japan) for supplying the β-arrestin1/2 KO cell line. Frederic Jean-Alphonse (CNRS, Nouzilly) for β-arrestin-YFP plasmids, Jonathan Javitch (Univ. Columbia, USA) for WT D$_2$R plasmids. We would like to acknowledge the use of Biorender.com for the production of schematic figures. This project was funded by Biotechnology and Biological Sciences Research Council (Grant BB/M011178/1).

## Author contributions

B.B. and A.C.H. conceived the study and, with K.L.S., designed the experiments. K.L.S performed the majority of wet laboratory experiments, analysis and figure preparation with contributions from Y.L for certain βarr BRET recruitment assays and molecular biology; A.J.M. for certain cAMP assays; W.Y for some confocal microscopy imaging. F.F. performed all molecular modeling and analysis. K.L.S wrote the majority of the manuscript with A.C.H and B.B. F.F wrote the sections of the manuscript relating to molecular modeling, with inputs and review by K.L.S, B.B., and A.C.H. All authors contributed to and approved the final manuscript.

## Competing interests

The authors declare no competing interests.
