## [Transparent Peer Review file · Nature Communications]

Stabilized D₂R G protein-coupled receptor oligomers identify multi-state β -arrestin complexes

Corresponding Author: Professor Aylin Hanyaloglu

Version 0:

Reviewer comments:

Reviewer #1

(Remarks to the Author)

In this manuscript, Sharrocks et al. identified residues located from D2R involved in the formation of D2R homodimers. These residues were mutated to stabilise and promote D2R homodimer formation. The corresponding mutant D2 receptors were characterised and compared to wild-type D2R for their respective ability to engage Gi, β -arrestin and ERK1/2 signalling as well as their trafficking properties. This work builds on an extensive list of GPCRs reported to form homo/hetero-dimers. Although the concept and existence of GPCR dimers is still controversial and highly debated, their presence (or potential presence) represents an additional way for these receptors to fine-tune and expand the repertoire of downstream cellular responses reminiscent of the complexity of the physiology. This work is of potential significance in the field of neurological diseases as D2R is primarily involved in locomotion, Parkinson's disease and schizophrenia. Although highly interesting, this work would benefit from major revision as in many situations cited below, additional or stronger evidence would significantly improve the robustness of the data and therefore the accuracy of the interpretation, claims and conclusions. Here is my list of major and minor concerns:

MAJOR CONCERNS:

- Under non-reducing conditions, all receptors should be in dimeric complexes (>180 kDa). However, the contrast of the western blot represented in panel 1b seems higher compared to the contrast of panel 1c. Consequently, it is difficult to objectively determine if the absence of any monomeric form in panel 1b wouldn't be due to an increase of the contrast that would artificially make the monomeric bands less visible. Would it be possible for the authors to improve the quality of this figure to show a western blot for panel 1b with the same contrast as the blot shown on panel 1c? The amount of proteins loaded on the gel on panel 1b is also very variable between wells (according to the tubulin control). A lower amount of total proteins loaded would also result in less visible monomeric bands. Would an alternative image of this gel be more representative from the western blots performed?
- Surprisingly, in the supplementary figure 1, there is a major difference between the BRETmax values in HEK293 cells and in β -arrestin1/2 KO cells when comparing the same receptors and same treatments. The higher BRETmax values in β -arrestin1/2 KO cells may potentially suggest that the absence of arrestins increases the formation of D2R dimers. In other words, arrestins seem to favour the monomeric state. Would it be possible for the authors to explain this observation as it would intuitively go against the general idea of the paper? Any alternative explanation of the mechanism underlying this observation?
- Although the use of PD-PALM with TIRF imaging allows the visualisation of individual receptors, it does not allow unfortunately to distinguish between real specific interaction and simply random proximity due to random collision. The differences in % of receptors in dimers although significant are also very minimal in figure 2. Surprisingly, while the proportion of receptors in di/oligomer increases for the V96S/V97C double mutant in basal state compared to the wild-type D2 receptor, there is no statistical difference between wild-type and mutant receptors upon agonist treatment (panel 2e). This contradicts the absence of effect of agonist treatment on receptor dimerization observed in supplementary figure 1. How the authors would explain this apparent discrepancy?
- When comparing the agonist-induced reduction of cAMP production by the mutant D2R compared to the wild-type D2 receptor (figure 3a), it seems that all receptors equally reduce cAMP. The only visible difference is the basal activity of each

receptor. Would it be possible to represent the dose-response curves from figure 3a as % of change from basal so all curves start at 100? This way, the agonist-mediated response would be clearer and distinctly represented from the basal activity. I am not convinced that there is a notable difference in the extent of agonist-mediated response of any mutant receptor compared to wild-type. How did the authors calculate their data in figure 3b? It is also important to mention that any difference in receptor responsiveness observed could be artificially due to differences in receptor expression at the plasma membrane. Did the authors measure wild-type and mutant receptor expression in parallel in these sets of experiments? Assessment of receptor activity should be performed at equivalent (matched) receptor expression. I also have the same concerns for the measurement of β -arrestin recruitment to the receptors (figure 3j-l). The basal needs to be subtracted to assess the agonist-induced response. The respective receptor expression should also be assessed when measuring β -arrestin recruitment.

- The data representing the kinetic of β -arrestin recruitment to each receptor (supplementary figure 6) are unfortunately not robust enough (at least in my opinion) to obtain a precise value of halftime and therefore conclude that V96C recruits β -arrestin faster than wild-type receptor.

- The authors used flow cytometry to compare the relative expression at plasma membrane of the mutant receptors (figure 4a). Unfortunately the variation between experiments seems quite important. Would it be possible for the authors to see if normalisation as % of wild-type would reduce this variation? The robustness of the data are important to get accurate statistical analyses. Alternatively, maybe using a conventional ELISA would work better than FACS? If the expression of V96S/V97C is indeed lower than wild-type at plasma membrane as claimed by the authors, it implies that the agonist-induced decrease of cAMP and beta-arrestin recruitment were also probably measured at different receptor levels. This is quite concerning.

- I am also concerned about the variability obtained for the activation of ERK1/2 by mutant and wild-type receptors (figure 5). The use of an alternative method such as AlphaScreen technology would more robustly assess ERK1/2 phosphorylation and strengthen the interpretations and conclusions.

MINOR CONCERNS:

- For the calculation of the dimer/monomer ratios on panel 1d, the authors did not specify which band was used for the monomeric state (110 kDa, 90 kDa, or both). Please, could the authors specify this in the figure legend?

- In the supplementary figure 1, the changes of BRETmax and BRET50 are represented as fold difference from basal and the statistical analyses are performed on this parameter. Why did the authors not represent the data as BRETmax and BRET50 as on figure 1 (i-n) for consistency purpose?

Reviewer #2

(Remarks to the Author)

Sharrocks et al have studied the oligomerisation of the dopamine D2R receptor and the role oligomers play in influencing receptor function. The authors use modelling, mutagenesis and single molecule microscopy among other techniques to support the claim that oligomers can be altered by point mutations and that by driving more oligomers they can alter signalling pathways, including recruitment of beta-arrestin2.

These claims in my opinion are not backed by enough convincing evidence. The effects they see are too small to be convincing. The distribution and lack of significance in several of the graphs of the BRET data in figure 1 suggests that there is not a big change in oligomerisation. Have the authors tried more controls to convince? What happens with obligate dimer class C in these experiments? What about comparing against a receptor known to not form dimers? There are several reports of arrestin biased D2Rs, why not try the published biased D2R receptors in these assays? Eg. Marc Caron's work For the V96S and V97C can they talk about the distances between the two protomers / amino acids? It is unclear whether a disulfide would bridge that gap. Disulfides are usually only 2-3Å, this gap looks larger but it is hard to tell.

The microscopy data appears more predictable and precise but the differences are even lower here between the constructs, supporting that the hypothesis just doesn't seem supported.

In the functional data the differences again appear barely significant with large distributions among experiments for the double mutant that look like the data is skewed. The authors suggest that there are altered arrestin and Gai signalling but they don't plot a web plot or try to calculate any bias. It might help to convince the reader. The most differences I see are in the 20% range at most which in my mind is within experimental error (Fig 3j) I don't quite understand what is plotted in k and l by comparison. It seems it's a single concentration point from the response curve in j.

The trafficking data in Fig 4 is interesting and perhaps the most convincing data but makes me wonder how these changes are achieved and if we can really learn anything about WT function from these. What is the connection to oligomerisation? That oligomers drive internalization?

The pERK1/2 data is weak. There are better ways to quantify pERK1/2 including HTRF assays and biosensors. These data do not convince and seem very subjective.

The authors put a model that suggests weaker G-protein binding / recruitment, yet they don't appear to have looked at this at all? It seems to me a G-protein activation assay would be required to support that model beyond what they have done.

All in all this study needs much more convincing data to support what is proposed.

Reviewer #3

(Remarks to the Author)

The manuscript "Stabilised D2R G-protein coupled receptor oligomers identify multi-state β -arrestin complexes" by Sharrocks et al. describes the role of oligomerization of D2R in arr2 -biased signaling. The authors engineered stable homodimers by altering residues within the H1-H2 interface and found that the D2R mutant homodimers are biased towards arr2 recruitment. While some of the experiments in the manuscript seem well-established, I have concerns regarding the use of ligands and the validity of the oligomers. In general, the manuscript is difficult to follow in its current form. Below are the major and minor comments on the manuscript.

Major comments:

- (1) My main concern with the manuscript is that the conclusions are made based on data from only agonist quinpirole; any experiments with antagonist, partial agonist, or second agonist would help support the conclusions made by the manuscripts.
- (2) I don't see clear evidence of the formation of oligomers from the Western blots. These blots show dimers with or without glycosylated, but not higher oligomers. Also, the quality of the blots is not impressive. Is it true that the expression of some mutants (V96S and Y93C) is lower? Or is this just a loading artifact similar to loading control -tubulin.
- (3) The authors mention in the discussion section that there is a '~6% increase' in an oligomeric state in the absence of a ligand. But on page 9, the authors mentioned ~17% of dimers and a range of higher-order receptor oligomers. It is not clear the percentage of receptors in dimeric and higher ordered oligomers or even protein aggregates due to misfolding. Super-resolution single-molecule experiments were performed using CAGE 500 dye-conjugated anti-FLAG antibody. It is not clear how many dyes were conjugated per antibody and how that affect the counting of receptors. A single molecule photobleaching experiment using FLAG and fluorescence protein (Venus?) tagged will help count monomers, dimers, and higher-ordered oligomers more efficiently.
- (4) Figure 3: At least 3 different concentrations (100 nM, 1 mM, and 10 nM) of quinpirole were used. The context of using different concentrations was not clear in the text and figure legends.

Other comments:

- (1) The citation of reference #9 is not clear.
- (2) Figure 1 a: labeling helices increases the clarity and makes it easy to follow.
- (3) Figure 1 e: Was the YFP attached to D2R?
- (4) A table with a list of all the plasmids and constructs will help.
- (5) Figures 1f, g, and h: Missing units, why are the number of points different for different samples? For example, in f, the WT has fewer points than the V96C. This might affect the saturation points when calculating BRET50 values.
- (6) Page 10: The sentence "In the absence of agonist, no change in the trends between WT and mutant D2Rs due to receptor density were observed" is not clear.
- (7) Figure 2, specifically d and f, are hard to follow. Increasing the bar width and changing color patterns may help.
- (8) It is not clear how figures 4a, b, and c are generated.
- (9) Was T4L included while modeling from 6CM4?
- (10) Are these mutants form heterodimers with A2AAR?

Reviewer #4

(Remarks to the Author)

Sharrocks et.al studied the impact of GPCR dimers on biased GPCR signaling. They worked with D₂R homodimer, which is famous for its contribution to the onset of various neurological disorders. They showed that D₂R homodimer prefers signaling through β -Arrestin 2, which was guided by computational and verified by experimental studies. The manuscript is well-written, and the results are sound. It is of significance to researchers in the field of including but not limited to biochemists, pharmacologists, biophysicists and molecular biologists.

My concern was that the authors did not provide any background information regarding the selection of the activation states of the protomers in the dimer in modeling studies. They used the asymmetric D₂R homodimer, where one of the protomers was active, while the other was inactive. This concept was introduced in Han et.al study in 2009, where the authors showed that maximum G protein signaling was achieved from D₂R homodimer when the dimer was asymmetric. Here, the authors showed a similar result for Arrestin signaling as well. Therefore, I think citing this study will strengthen their discussion.

Also, since the authors commented on the interaction interface formed between the D₂R homodimer and β -Arrestin 2, they should perform atomistic MD simulations using the modeled complex in the membrane environment, preferably with composition that mimics lipid rafts (including PIP2, cholesterol, etc), to prove the stability of the complex and reliability of the interactions reported.

They should also provide information on the sequence similarities of GPCRs which were used as templates in modeling studies.

Corresponding reference: Han Y, Moreira IS, Urizar E, Weinstein H, Javitch JA. Allosteric communication between protomers of dopamine class A GPCR dimers modulates activation. *Nature Chemical Biology*. 2009 Sep;5(9):688-695

Version 1:

Reviewer comments:

Reviewer #1

(Remarks to the Author)

The authors have addressed very well all my concerns. I am happy with the updated version of the manuscript.

(Remarks on code availability)

Reviewer #3

(Remarks to the Author)

The authors have addressed most of my comments in the revised manuscript.

(Remarks on code availability)

Reviewer #4

(Remarks to the Author)

The authors addressed the questions raised. The current version is suitable for publication.

(Remarks on code availability)

Reviewer #5

(Remarks to the Author)

Sharrocks et al report an interesting study aimed at linking GPCR oligomerization and modulation of transducer coupling properties. Overall, I find the experiments to be technically solid and to include some interesting observations pointing toward D2R dimerization boosting arrestin coupling relative to G protein coupling. However, I do find the experiments relatively indirect, especially the proposed 2:2 D2R:arrestin model, and somewhat over-interpreted. I have some minor suggestions to tone down some key points and to add a little bit more context from the literature:

-In the introduction, it is worth acknowledging Moller et al, *Nat Chem Bio*, 2020 as this is an interesting study proposing that arrestin can drive dimerization of the MOR.

-Overall there is not sufficient acknowledgement of the role of GRKs. Experiments could address if there is a difference in phosphorylation between mutants or KO cells could be used to assess if GRK-independent arrestin-coupling is altered. At least the potential for GRKs to be driving the ultimate effects on arrestin should be stated somewhere.

-In addition, there should be some more discussion of the potential that mutations have an effect on TMD monomer conformational dynamics and that this could explain some of the functional effect.

-A more direct measurements of receptor/G protein and receptor/arrestin stoichiometry would be ideal. But, I appreciate that this is quite difficult. Perhaps the authors can more clearly acknowledge that this will eventually be needed to substantiate some of the claims.

-The variability in arrestin orientation across family A GPCR structures warrants a bit more discussion. While it is nice that two different orientations are accommodated in 2:2 models, it would be good to assess all known GPCR/arrestin angles and to see if there are any that provide clashes. It is worth noting that tail-only conformations may also contribute which could mitigate this issue.

-Along these lines, 2:2 GPCR/arrestin stoichiometries have been recently seen for family C GPCRs, mGluR3 and mGluR8 (et al, *Nat Chem Bio*, 2025; Marx et al, *bioRxiv*, 2025) and warrant a mention although it has not been established how the presence of 2 arrestins alters internalization or signaling properties.

(Remarks on code availability)

Reviewer #1

In this manuscript, Sharrocks et al. identified residues located from D2R involved in the formation of D2R homodimers. These residues were mutated to stabilise and promote D2R homodimer formation. The corresponding mutant D2 receptors were characterised and compared to wild-type D2R for their respective ability to engage Gi, β -arrestin and ERK1/2 signalling as well as their trafficking properties. This work builds on an extensive list of GPCRs reported to form homo/hetero-dimers. Although the concept and existence of GPCR dimers is still controversial and highly debated, their presence (or potential presence) represents an additional way for these receptors to fine-tune and expand the repertoire of downstream cellular responses reminiscent of the complexity of the physiology. This work is of potential significance in the field of neurological diseases as D2R is primarily involved in locomotion, Parkinson's disease and schizophrenia. Although highly interesting, this work would benefit from major revision as in many situations cited below, additional or stronger evidence would significantly improve the robustness of the data and therefore the accuracy of the interpretation, claims and conclusions. Here is my list of major and minor concerns:

Thank you for the supportive and constructive feedback. Below, we respond to each comment and describe the changes made.

MAJOR

CONCERNS:

1. Under non-reducing conditions, all receptors should be in dimeric complexes (>180 kDa). However, the contrast of the western blot represented in panel 1b seems higher compared to the contrast of panel 1c. Consequently, it is difficult to objectively determine if the absence of any monomeric form in panel 1b wouldn't be due to an increase of the contrast that would artificially make the monomeric bands less visible. Would it be possible for the authors to improve the quality of this figure to show a western blot for panel 1b with the same contrast as the blot shown on panel 1c? The amount of proteins loaded on the gel on panel 1b is also very variable between wells (according to the tubulin control). A lower amount of total proteins loaded would also result in less visible monomeric bands. Would an alternative image of this gel be more representative from the western blots performed?

We have avoided modulation of western blot images, however different exposures of the image shown in Fig. 1b (now supplementary Fig. 1) do not change the visibility of the bands detected i.e. minimal monomeric bands for the mutant are detected (please see Response to Reviewers Fig. 1 below). We understand the reviewer's concern about variable loading, however, we did not quantify dimer levels from the Western blot analysis in non-reducing conditions, nor do we make any statements about the quantity of dimer for the mutant vs WT in non-reducing conditions in Supplementary Fig. 1 (previously Fig. 1b). The only observation from supplementary Fig. 1 is that the majority of the receptor appears in a higher order form. The primary finding from these Western blots is that there is less monomer detectable for the mutant compared to WT under reducing conditions (now Fig. 1b-c in revised manuscript). As requested by the reviewer, we have also provided an alternative Western blot (revised Supplementary Fig.1). To improve the clarity of our conclusions within the manuscript

from these Western blots, we have moved the non-reducing Western blot (previously Fig. 1b) to Supplementary Fig. 1 to emphasise that the statements we make in the text and the quantification are regarding the Western blot carried out specifically under reducing conditions. We have modified the text on page 7 of the manuscript to increase clarity of both the results and their interpretation.

Response to reviewers Fig.1. Alternative contrast for Western blot presented in Supplementary Fig. 1a. Western blot of HEK293 cell lysates transfected with D₂R-Rluc8 WT or mutant constructs carried out under non-reducing conditions. Probed with M1 anti-FLAG antibody to detect FLAG-tagged receptor and anti- α -tubulin as a loading control. The same western blot image (a) was altered to reduce the contrast of the image (b) to match the contrast levels of the Western blot that was carried out under reducing conditions (Fig. 1b).

2. Surprisingly, in the supplementary figure 1, there is a major difference between the BRET_{max} values in HEK293 cells and in β -arrestin1/2 KO cells when comparing the same receptors and same treatments. The higher BRET_{max} values in β -arrestin1/2 KO cells may potentially suggest that the absence of arrestins increases the formation of D₂R dimers. In other words, arrestins seem to favour the monomeric state. Would it be possible for the authors to explain this observation as it would intuitively go against the general idea of the paper? Any alternative explanation of the mechanism underlying this observation?

Thank you for this suggestion. The data that was previously presented in Supplementary Fig. 1 was basal and ligand-induced changes in BRET_{max} and BRET₅₀, to allow comparisons between cell lines, which are not significantly different. The prior data on comparing fold change values across cell lines was confusing given there were no significant differences between basal and agonist induced BRET signals within a cell lines. Therefore, we have modified this figure and removed the comparison, including any conclusions that arrestin modifies the interaction, rather that the conformational changes induced by select mutants favoring a more stable or enhanced association, alters arrestin interactions and functions in distinct manners. Supplementary Fig. 14 c and d show the BRET_{max} and BRET₅₀ data before and after quinpirole treatment and there is no significant difference between BRET_{max} in WT HEK293 and β arr 1/2 KO D₂R WT or V96C expressing cells, suggesting no significant increase in protomer proximity in the absence of β arr 1/2.

Additionally, we welcome the suggestion proposed by the reviewer in point 9 to present the data as raw BRETmax and BRET50 under basal and agonist stimulated conditions to aid the interpretation of this data for readers and have presented this in the revised manuscript as part of Supplementary Figure 3 and 14.

3. Although the use of PD-PALM with TIRF imaging allows the visualisation of individual receptors, it does not allow unfortunately to distinguish between real specific interaction and simply random proximity due to random collision. The differences in % of receptors in dimers although significant are also very minimal in figure 2. Surprisingly, while the proportion of receptors in di/oligomer increases for the V96S/V97C double mutant in basal state compared to the wild-type D2 receptor, there is no statistical difference between wild-type and mutant receptors upon agonist treatment (panel 2e). This contradicts the absence of effect of agonist treatment on receptor dimerization observed in supplementary figure 1. How the authors would explain this apparent discrepancy?

We respectfully disagree. Our group has published extensively with this methodology, providing detailed protocols, including numerous distinct controls. These include membrane proteins that do not interact, randomised x,y coordinate datasets for the near neighbourhood analysis and mutational disruption of the dimer interaction¹⁻⁴. Additionally, in this study, as reported in other studies, di/oligomerisation is detected with similar profiles across a broad range of receptor expression levels (supplementary Fig 5). Although our data is also in line with other studies assessing D2R dimerization, we do acknowledge that PD-PALM provides spatial resolution at the expense of temporal resolution. We would like to highlight that our prior experience across a range of Class A and Class B GPCRs demonstrates that while the profiles of complexes can vary and be receptor dependent, 'small' changes in reorganisation quantified by PD-PALM has a profound impact on the overall function. For example, ligand induced changes specifically in the protomer composition of oxytocin receptor/prostaglandinEP2 heterotetramers, underly the ability of the complexes to modulate G protein coupling and signalling^{2,4}

For the revised manuscript, we have added further explanation on page 11 where we highlight that while PD-PALM microscopy allows quantification of D2R di/oligomers at the cell surface, BRET assays are unable to distinguish between plasma-membrane receptor and intracellular receptors (both biosynthetic and endocytic). We now state that the apparent discrepancies in agonist effect on dimerization between BRET and PD-PALM experiments may suggest that this is a result of BRET assays detecting interactions between receptors located across subcellular compartments (now supplementary Fig.3 and 5 and Fig. 2e).

4. When comparing the agonist-induced reduction of cAMP production by the mutant D2R compared to the wild-type D2 receptor (figure 3a), it seems that all receptors equally reduce cAMP. The only visible difference is the basal activity of each receptor. Would it be possible to represent the dose-response curves from figure 3a as % of change from basal so all curves start at 100? This way, the agonist-mediated response would be clearer and distinctly represented from the basal activity. I am not convinced that there is a notable difference in the extent of agonist-mediated response of any mutant receptor compared to wild-type. How did the authors calculate their data in figure 3b? It is also important to mention that any difference in receptor

responsiveness observed could be artificially due to differences in receptor expression at the plasma membrane. Did the authors measure wild-type and mutant receptor expression in parallel in these sets of experiments? Assessment of receptor activity should be performed at equivalent (matched) receptor expression. I also have the same concerns for the measurement of β -arrestin recruitment to the receptors (figure 3j-l). The basal needs to be subtracted to assess the agonist-induced response. The respective receptor expression should also be assessed when measuring β -arrestin recruitment.

We recognise the reviewers concerns about receptor expression levels, however these are addressed by the data presented now in Supplemental Fig. 2e-h demonstrating that the total receptor expression is not significantly different across Wt and mutant receptors and with control experiments with lower cell surface WT D2R expression (Supplementary Fig. 9). Given that decreasing cell surface WT D2R to a similar level as the V96S/V97C receptor, does not result in the same signalling changes in cAMP inhibition, β -arrestin recruitment and internalization that are observed in mutant receptors, we concluded that the functional effects observed with mutant receptors are due to the specific mutation and not the result of lower expression levels. This is stated on page 20 and in the revised manuscript, has now been highlighted earlier in the manuscript when describing the cAMP and β -arrestin recruitment assays.

We have clarified the method for how cAMP levels are calculated in the figure legend and present the data normalized to 100%, to address the reviewers concerns about Fig. 3a and b. The kinetics of β -arrestin recruitment were calculated from ligand-induced BRET profiles (Supplementary fig 8), with basal levels subtracted, as the reviewer suggests. We have ensured this is now clear in the text.

5. The data representing the kinetic of β -arrestin recruitment to each receptor (supplementary figure 6) are unfortunately not robust enough (at least in my opinion) to obtain a precise value of halftime and therefore conclude that V96C recruits β -arrestin faster than wild-type receptor.

As with any biosensor, we acknowledge the limitations of the BRET-based assay for arrestin/receptor binding kinetics, although this technique is employed widely in the field. We have mutations that either impact or do not impact recruitment profiles, and exhibit dose-dependent profiles in line with prior published data for the WT receptor.

The kinetic values represent additional information from the BRET profiles, which demonstrate that the V96C mutant has an altered recruitment profile and V96S/V97C exhibits higher basal arrestin recruitment. We have altered the text to ensure the data described is not overstated and have included text in the discussion stating that such interactions, even if sustained in profile, may represent multiple interactions of the GPCR with distinct arrestin conformations, as recently described by Grimes et al., Cell, 2023.

6. The authors used flow cytometry to compare the relative expression at plasma membrane of the mutant receptors (figure 4a). Unfortunately the variation between experiments seems quite important. Would it be possible for the authors to see if normalisation as % of wild-type would reduce this variation? The robustness of the data are important to get accurate statistical analyses. Alternatively, maybe using a

conventional ELISA would work better than FACS? If the expression of V96S/V97C is indeed lower than wild-type at plasma membrane as claimed by the authors, it implies that the agonist-induced decrease of cAMP and beta-arrestin recruitment were also probably measured at different receptor levels. This is quite concerning.

As discussed in our response to point 4, the experiments in Supplementary Fig. 9 suggest lower surface expression levels are not an underlying reason for the changes observed in the cAMP and b-arrestin recruitment assays with the mutant receptors. We can present the data as a percentage of WT as the reviewer suggests, however this would not alter the conclusion as V96S/V97C expression is lower at the plasma membrane due to increased basal internalization, but does not have significantly lower total expression (as shown by D2R-Venus fluorescence values in Supplementary Fig.2). The initial reason for investigating receptor internalisation was indeed this interesting observation that some receptors had lower plasma membrane expression levels.

7. I am also concerned about the variability obtained for the activation of ERK1/2 by mutant and wild-type receptors (figure 5). The use of an alternative method such as AlphaScreen technology would more robustly assess ERK1/2 phosphorylation and strengthen the interpretations and conclusions.

As suggested we repeated these assays investigating quinpirole-induced ERK activation using a HTRF based phosphoERK assay (Revvity). However, **we found the assay window of this method to be limited** compared to the data we obtain from western blots. A fold change of between 1 and 2 (with two fold being the max for this system with the manufacturer's supplied positive control) is observed for the HTRF based assay, while we obtained average fold changes of up to 8 in western blot assays (please see Response to reviewers Fig. 2 below and manuscript Fig.5). Therefore, we believe that this assay reaches a maximum detection level, which means it does not adequately capture the large changes in phospho-ERK activation that we observe for D2R mutants. To illustrate this, we ran the same lysates used in the HTRF assay on a Western blot, probed for pERK and total ERK and quantified the relative phospho-ERK levels (pERK/total ERK) as fold change from basal (Response to reviewers Fig. 2). In addition, this Western blot gave us additional biological replicates, which has been added to the data shown in Fig. 5 b and c. For data transparency, we have also included the raw phospho-ERK/total ERK data (not presented as a fold change from basal) in Supplementary Fig. 12. When not normalised to basal, a similar trend is observed, with an increase in ligand induced phospho-ERK levels observed in D2R V96C transfected HEK293 cells compared to WT. The same trends are observed in each biological repeat, but with variation in the extent of these differences. Given these observations are reproducible across multiple biological replicates, we believe this adequately demonstrates the robustness of these findings (see source data file for Fig. 5). In the revised manuscript we have modified the scale of the y axis in Fig 5c to be consistent with Fig 5b so differences between WT HEK293 cells and β arr 1/2 knockout cells are more obvious. We have also added a side-by-side comparison of WT HEK293 cells and β arr 1/2 knockout cells expressing D2R V96C, V96S and v96S/V97C in Fig 5. f-h.

Response to reviewers Fig. 2. Comparison of western blot and HTRF-based phospho-ERK assay. HEK293 cells (a,b,c) or β arr 1/2 knockout (KO) cells (d,e,f) were transfected with D₂R WT (black), V96C (blue), V96S (orange) or V96S/V97C (turquoise) and stimulated with 10 μ M quinpirole for 0 (basal), 2, 5, 10 or 15 minutes prior to cell lysis. The cell lysates were analysed by western blot (a and d) and quantified as phospho-ERK/total ERK as a fold change from basal (b and e). The phospho-ERK levels in the same cell lysates were determined by a HTRF based Phospho -ERK Advanced Detection Kit (Revity) assay (c and f) and normalised to total protein levels. N=1 shown here for side by side comparison with western blot, N=3 done in total.

MINOR

CONCERNS:

8. For the calculation of the dimer/monomer ratios on panel 1d, the authors did not specify which band was used for the monomeric state (110 kDa, 90 kDa, or both). Please, could the authors specify this in the figure legend?

The legend has been modified to include this information.

9. In the supplementary figure 1, the changes of BRETmax and BRET50 are represented as fold difference from basal and the statistical analyses are performed on this parameter. Why did the authors not represent the data as BRETmax and BRET50 as on figure 1 (i-n) for consistency purpose?

We have modified how the data is presented as the reviewer suggests- see above point 2.

Reviewer #2 (Remarks to the Author):

Sharrocks et al have studied the oligomerisation of the dopamine D2R receptor and the role oligomers play in influencing receptor function. The authors use modelling, mutagenesis and single molecule microscopy among other techniques to support the claim that oligomers can be altered by point mutations and that by driving more oligomers they can alter signalling pathways, including recruitment of beta-arrestin2.

1. These claims in my opinion are not backed by enough convincing evidence. The effects they see are too small to be convincing. The distribution and lack of significance in several of the graphs of the BRET data in figure 1 suggests that there is not a big change in oligomerisation. Have the authors tried more controls to convince? What happens with obligate dimer class C in these experiments? What about comparing against a receptor known to not form dimers? There are several reports of arrestin biased D2Rs, why not try the published biased D2R receptors in these assays? Eg. Marc Caron's work.

Please also see response to Point 3 from Reviewer 1. All experimental approaches are established platforms that have been applied across a range of GPCR dimers (homomers and heteromers) including mutations that disrupt these interactions or introduce an 'obligate functional dimer' for a Class A GPCR by the authors over studies from the past 15 years^{1,2,4-6}. The goal of this study was not to dissociate but to initially attempt to stabilise a potential Class A homodimer interaction, using the D2 dopamine receptor. Serendipitously, this revealed an insight into the complexity of conformations a GPCR can undertake, in particular impacting β arr2 associations and functions.

Comparing BRET profiles obtained from an obligate dimer and a solely monomeric receptor that would remain so throughout its biosynthetic, cell surface and endocytic journey as a control (if this were possible to identify) for all BRET assays for all receptors would not be a meaningful control in our opinion.

In this revised manuscript we have included a negative control within our BRET saturation assays (supplementary Fig 2 a). We have additional data from the D2R Y93C mutant that demonstrate there is no difference in BRETmax compared to WT and have included this in the revised supplementary data to support mutant specific changes (Supplementary Fig. 2 b-d). We demonstrate in all other experiments

assessing dimer stability and signalling that the D2R Y93C mutant is no different to WT. However, the mutants in which we observe changes in BRETmax compared to WT also show altered signalling profiles and/or increased receptor homomerization in other assays in this manuscript. We hope this provides additional robustness of the changes in BRETmax observed for the other mutant receptors.

Also, we should highlight that our findings across BRET assays, western blot and PALM all suggest that V96S/V97C increases dimer stability. We also provide additional new data with a β -arrestin biased D2R-selective ligand. The above discussions are highlighted in the text in the revised manuscript.

2. For the V96S and V97C can they talk about the distances between the two protomers / amino acids? It is unclear whether a disulfide would bridge that gap. Disulfides are usually only 2-3Å, this gap looks larger but it is hard to tell. In the revised legend to Fig. 6 we report the distance between the two sulfur atoms of C97 in the V96S/V97C mutant (2 Å).

3. The microscopy data appears more predictable and precise but the differences are even lower here between the constructs, supporting that the hypothesis just doesn't seem supported.

We respectfully disagree. Our group has published extensively with this methodology, providing detailed protocols, including numerous distinct controls. These include membrane proteins that do not interact, randomised x,y coordinate datasets for the near neighbourhood analysis and mutational disruption of the dimer interaction¹⁻⁴. Additionally, in this study, as reported in other studies, di/oligomerisation is detected with similar profiles across a broad range of receptor expression levels (supplementary Fig 5). Although our data is also in line with other studies assessing D2R dimerization, we do acknowledge that PD-PALM provides spatial resolution at the expense of temporal resolution. We would like to highlight that our prior experience across a range of Class A and Class B GPCRs demonstrates that while the profiles of complexes can vary and be receptor dependent, 'small' changes in reorganisation quantified by PD-PALM has a profound impact on the overall function. For example, ligand induced changes specifically in the protomer composition of oxytocin receptor/prostaglandinEP2 heterotetramers, underly the ability of the complexes to modulate G protein coupling and signalling^{2,4}

For the revised manuscript, we have added further explanation on page 11 where we highlight that while PD-PALM microscopy allows quantification of D2R di/oligomers at the cell surface, BRET assays are unable to distinguish between plasma-membrane receptor and intracellular receptors (both biosynthetic and endocytic). We now state that the apparent discrepancies in agonist effect on dimerization between BRET and PD-PALM experiments may suggest that this is a result of BRET assays detecting interactions between receptors located across subcellular compartments (now supplementary Fig.3 and 5 and Fig. 2e).

4. In the functional data the differences again appear barely significant with large distributions among experiments for the double mutant that look like the data is skewed. The authors suggest that there are altered arrestin and Gai signalling but they don't plot a web plot or try to calculate any bias. It might help to convince the reader.

The most differences I see are in the 20% range at most which in my mind is within experimental error (Fig 3j) I don't quite understand what is plotted in k and l by comparison. It seems it's a single concentration point from the response curve in j. All the data was analysed with appropriate statistical tests to determine significance. We would like to highlight that the critical observation here is the increased basal activity of V96S/V97C in terms of arrestin recruitment and arrestin-mediated functions. A web plot has been provided in the revised manuscript in Fig. 6. Additionally, we have provided further clarification in the legend for Figs. 3k and 3l.

5. The trafficking data in Fig 4 is interesting and perhaps the most convincing data but makes me wonder how these changes are achieved and if we can really learn anything about WT function from these. What is the connection to oligomerisation? That oligomers drive internalization?

The mutations that exhibit altered arrestin recruitment profiles lead to altered internalization. We propose that GPCRs exhibit multiple conformations, and that these modifications of D2R highlight conformational forms of the WT receptor, even if transient, which would promote distinct arrestin-mediated functions such as internalization or ERK signalling.

6. The pERK1/2 data is weak. There are better ways to quantify pERK1/2 including HTRF assays and biosensors. These data do not convince and seem very subjective. As suggested we repeated these assays investigating quinpirole-induced ERK activation using a HTRF based phosphoERK assay (Revvity). However, **we found the assay window of this method to be limited** compared to the data we obtain from western blots. A fold change of between 1 and 2 (with two fold being the max for this system with the manufacturer's supplied positive control) is observed for the HTRF based assay, while we obtained average fold changes of up to 8 in western blot assays (please see Response to reviewers Fig. 2 below and manuscript Fig.5). Therefore, we believe that this assay reaches a maximum detection level, which means it does not adequately capture the large changes in phospho-ERK activation that we observe for D2R mutants. To illustrate this, we ran the same lysates used in the HTRF assay on a Western blot, probed for pERK and total ERK and quantified the relative phospho-ERK levels (pERK/total ERK) as fold change from basal (Response to reviewers Fig. 2). In addition, this Western blot gave us additional biological replicates, which has been added to the data shown in Fig. 5 b and c. For data transparency, we have also included the raw phospho-ERK/total ERK data (not presented as a fold change from basal) in Supplementary Fig. 12. When not normalised to basal, a similar trend is observed, with an increase in ligand induced phospho-ERK levels observed in D2R V96C transfected HEK293 cells compared to WT. The same trends are observed in each biological repeat, but with variation in the extent of these differences. Given these observations are reproducible across multiple biological replicates, we believe this adequately demonstrates the robustness of these findings (see source data file for Fig. 5). In the revised manuscript we have modified the scale of the y axis in Fig 5c to be consistent with Fig 5b so differences between WT HEK293 cells and β arr 1/2 knockout cells are more obvious. We have also added a side-by-side comparison of WT HEK293 cells and β arr 1/2 knockout cells expressing D2R V96C, V96S and v96S/V97C in Fig 5. f-h.

Response to reviewers Fig. 2. Comparison of western blot and HTRF-based phospho-ERK assay. HEK293 cells (a,b,c) or β arr 1/2 knockout (KO) cells (d,e,f) were transfected with D₂R WT (black), V96C (blue), V96S (orange) or V96S/V97C (turquoise) and stimulated with 10 μ M quinpirole for 0 (basal), 2, 5, 10 or 15 minutes prior to cell lysis. The cell lysates were analysed by western blot (a and d) and quantified as phospho-ERK/total ERK as a fold change from basal (b and e). The phospho-ERK levels in the same cell lysates were determined by a HTRF based Phospho -ERK Advanced Detection Kit (Revity) assay (c and f) and normalised to total protein levels. N=1 shown here for side by side comparison with western blot, N=3 done in total.

7. The authors put a model that suggests weaker G-protein binding / recruitment, yet they don't appear to have looked at this at all? It seems to me a G-protein activation assay would be required to support that model beyond what they have done.

We do not make any conclusions about the affinity/binding of G-proteins to the dimer in our model. We are suggesting a relationship whereby there would be fewer G-proteins available for signalling when D2R is in the dimeric form, compared to the two β -arrestins per dimer proposed by computational modelling, which may provide an explanation for the biased signalling.

All in all this study needs much more convincing data to support what is proposed.

We take on board the queries/concerns raised by the reviewer and have welcomed the opportunity guided by the reviewer to improve the clarity and robustness of our data, as detailed above.

Reviewer #3 (Remarks to the Author)

The manuscript "Stabilised D2R G-protein coupled receptor oligomers identify multi-state β -arrestin complexes" by Sharrocks et al. describes the role of oligomerization of D2R in barr2-biased signaling. The authors engineered stable homodimers by altering residues within the H1-H2 interface and found that the D2R mutant homodimers are biased towards barr2 recruitment. While some of the experiments in the manuscript seem well-established, I have concerns regarding the use of ligands and the validity of the oligomers. In general, the manuscript is difficult to follow in its current form. Below are the major and minor comments on the manuscript. We thank the reviewer for their positive and constructive feedback. Please find below detailed responses, including addition of new data.

Major comments:

(1) My main concern with the manuscript is that the conclusions are made based on data from only agonist quinpirole; any experiments with antagonist, partial agonist, or second agonist would help support the conclusions made by the manuscripts.

We acknowledge that it is important to assess the effects of additional ligands. For this revised manuscript we have assessed the effect of UNC9994 (a β -arrestin biased agonist/partial agonist) in key experiments assessing dimerization and signaling (Supplementary Fig. 10). It is also important to note that the most significant impact of stabilising D2R is on the ligand-independent effects. We feel that analysis with further ligands represents work that is beyond the scope of the current study.

(2) I don't see clear evidence of the formation of oligomers from the Western blots. These blots show dimers with or without glycosylated, but not higher oligomers. Also, the quality of the blots is not impressive. Is it true that the expression of some mutants (V96S and Y93C) is lower? Or is this just a loading artifact similar to loading control α -tubulin.

It is difficult to resolve higher-order transient oligomers by SDS-Page. SDS is highly likely to disrupt the weaker interactions that might be responsible for these oligomers. Additionally it is difficult to determine whether oligomers observed using this technique are artefactual due to heating of the sample during electrophoresis. Given we cannot confirm the presence of specific higher-order oligomers, we have undertaken quantification on dimeric receptors (shown in Fig. 1c. which we have clarified in the

text of the revised manuscript. Differences in expression levels are addressed in figure 4.

(3) The authors mention in the discussion section that there is a '~6% increase' in an oligomeric state in the absence of a ligand. But on page 9, the authors mentioned ~17% of dimers and a range of higher-order receptor oligomers. It is not clear the percentage of receptors in dimeric and higher ordered oligomers or even protein aggregates due to misfolding. Super-resolution single-molecule experiments were performed using CAGE 500 dye-conjugated anti-FLAG antibody. It is not clear how many dyes were conjugated per antibody and how that affect the counting of receptors. A single molecule photobleaching experiment using FLAG and fluorescence protein (Venus?) tagged will help count monomers, dimers, and higher-ordered oligomers more efficiently.

The 6% increase is the increase in all oligomer types in V96S/V97C-expressing cells compared to WT under basal conditions, this sentence has been clarified in the text.

We have also added further clarification in the methods section stating the antibody:dye ratio was 1:1 which allows specific quantification of receptors at the single molecule level, as described in our prior publications and methods articles on PD-PALM. Each single molecule method has its advantages and limitations. SMLM such as PD-PALM provides the complete landscape of GPCR organization at the plasma membrane and represents a robust technique we have validated across a range of receptors^{1-4,6}.

(4) Figure 3: At least 3 different concentrations (100 nM, 1 mM, and 10 nM) of quinpirole were used. The context of using different concentrations was not clear in the text and figure legends. These are either based on concentration response curves or used as a 'maximal response' to quantify and compare ligand-induced changes between WT mutant- we have now clarified this in the relevant figure legends.

Other comments:

(1) The citation of reference #9 is not clear. – That reference was an error and has now been deleted

(2) Figure 1 a: labeling helices increases the clarity and makes it easy to follow. Labels have been added to the figure in the revised manuscript

(3) Figure 1 e: Was the YFP attached to D2R? Yes, as depicted in Fig.1E and also clarified in Methods.

(4) A table with a list of all the plasmids and constructs will help. Thank you for the helpful suggestion, this has now been added.

(5) Figures 1f, g, and h: Missing units, why are the number of points different for different samples? For example, in f, the WT has fewer points than the V96C. This might affect the saturation points when calculating BRET50 values. The specific graph that the reviewer is referring to does not have a different number of points.

(6) Page 10: The sentence "In the absence of agonist, no change in the trends between WT and mutant D2Rs due to receptor density were observed" is not clear. Sentence has now been clarified- "in the absence of agonist, differences between WT and mutant receptors were the same regardless of receptor density"

(7) Figure 2, specifically d and f, are hard to follow. Increasing the bar width and changing color patterns may help.

Figure has been modified.

(8) It is not clear how figures 4a, b, and c are generated.

Clarification is given in figure legend.

(9) Was T4L included while modeling from 6CM4?

No, T4L was removed.

(10) Are these mutants form heterodimers with A2AAR?

No, these are D2R homodimers.

Reviewer #4 (Remarks to the Author)

Sharrocks et.al studied the impact of GPCR dimers on biased GPCR signaling. They worked with D₂R homodimer, which is famous for its contribution to the onset of various neurological disorders. They showed that D₂R homodimer prefers signaling through β-Arrestin 2, which was guided by computational and verified by experimental studies. The manuscript is well-written, and the results are sound. It is of significance to researchers in the field of including but not limited to biochemists, pharmacologists, biophysicists and molecular biologists.

Thank you for the supportive comments and helpful feedback.

1. My concern was that the authors did not provide any background information regarding the selection of the activation states of the protomers in the dimer in modeling studies. They used the asymmetric D₂R homodimer, where one of the protomers was active, while the other was inactive. This concept was introduced in Han et.al study in 2009, where the authors showed that maximum G protein signaling was achieved from D₂R homodimer when the dimer was asymmetric. Here, the authors showed a similar result for Arrestin signaling as well. Therefore, I think citing this study will strengthen their discussion.

The choice of the D₂R models employed in docking simulations depended on the availability of structures in the Protein Data bank (PDB). Symmetrical D₂R homodimers with both protomers either in the inactive or active states as well as inactive-active dimers were predicted. The following structures were probed as sources of D₂R protomers: a) inactive-state structures: 6CM4, 6LUQ, and 7DFP (i.e. D₂R in complex with risperidone, haloperidol, and spiperone, respectively), and b) active-state structures: 7JVR and 8IRS (i.e. the ternary complexes between heterotrimeric Gi and D₂R in complex with bromocriptine and ritigotine, respectively). The best results in terms of reliable docking solutions (according to the membrane topology indices, see Methods)) were achieved, for the active-state dimers, by using D₂R from 8IRS as a protomer, and, for the inactive-state dimers, by-using D₂R from 6CM4 as a protomer. No reliable solutions were achieved for the asymmetric dimers. Since the main architecture of the most reliable inactive-state and active-state dimers is the same, only the active-state dimers in their WT and mutant forms are shown here. Indeed,

they are the most reliable and suitable for modelling the interactions with G protein and arrestin. Consistently, the mutants, especially the double mutant, are constitutively active. Text has been added to the manuscript to clarify this point. In the revised manuscript we cite the work by Han and co-workers.

2. Also, since the authors commented on the interaction interface formed between the D₂R homodimer and β -Arrestin 2, they should perform atomistic MD simulations using the modeled complex in the membrane environment, preferably with composition that mimics lipid rafts (including PIP₂, cholesterol, etc), to prove the stability of the complex and reliability of the interactions reported.

To prove the energetic stability of a huge complex made of a dimeric receptor and two arrestin-3 molecules embedded in lipids and water, requires long time-scale simulations. Furthermore, a reliable energetic estimation of stability or of persistence of intermolecular interactions would require a complete model of the receptor. This is not feasible because the huge third intracellular loop of D₂R cannot be reliably modelled even in its short isoform. In D₂R the phosphorylation sites are contained in juxta-membrane portions of the third intracellular loop, which are also absent in the model. Although it has been reported that D₂R phosphorylation by receptor kinases is not required for arrestin recruitment, it might influence stability determinations.

In our study we discuss that, in general, the relevant core interactions that occur in arrestin-GPCR complexes involve the arrestin crest- and C-loops, and the second intracellular loop of the receptor, all present in our models. We proposed two alternative models of core interactions depending on the different conformation of the finger loop. Collectively, we are confident of the overall architecture of the complex, which allowed us to make hypotheses on the possible arrestin:receptor stoichiometry. This we think is sufficient in the context of the present study. Long time-scale MD simulations would not change the final message nor reliably address the stability issue for the reasons stated above.

3. They should also provide information on the sequence similarities of GPCRs which were used as templates in modeling studies

No comparative modelling has been done as we have used the structures of the D₂R itself both in the inactive and active states (see above and the Methods).

Corresponding reference: Han Y, Moreira IS, Urizar E, Weinstein H, Javitch JA. Allosteric communication between protomers of dopamine class A GPCR dimers modulates activation. *Nature Chemical Biology*. 2009 Sep;5(9):688-695

References-

1. Jonas, K. C., Fanelli, F., Huhtaniemi, I. T. & Hanyaloglu, A. C. Single Molecule Analysis of Functionally Asymmetric G Protein-coupled Receptor (GPCR) Oligomers Reveals Diverse Spatial and Structural Assemblies *Journal of Biological Chemistry* **290**, 3875–3892 (2015).
2. Jonas, K. C. *et al.* Temporal reprogramming of calcium signalling via crosstalk of gonadotrophin receptors that associate as functionally asymmetric heteromers. *Sci Rep* **8**, 2239 (2018).
3. Casarini, L. *et al.* Membrane Estrogen Receptor (GPER) and Follicle-Stimulating Hormone Receptor (FSHR) Heteromeric Complexes Promote Human Ovarian Follicle Survival. *iScience* **23**, (2020).
4. Walker, A. R. *et al.* Functional rewiring of G protein-coupled receptor signaling in human labor. *Cell Rep* **40**, (2022).
5. Jonas, K. C. & Hanyaloglu, A. C. Super-Resolution Imaging as a Method to Study GPCR Dimers and Higher-Order Oligomers. in *Receptor-Receptor Interactions in the Central Nervous System* (eds. FUXE, K. & Borroto-Escuela, D. O.) 329–343 (Springer New York, New York, NY, 2018). doi:10.1007/978-1-4939-8576-0_21.
6. Agwuegbo, U. C. & Jonas, K. C. Molecular and functional insights into gonadotropin hormone receptor dimerization and oligomerization. *Minerva Ginecol* **70**, (2018).

Reviewer #5 (Remarks to the Author):

Sharrocks et al report an interesting study aimed at linking GPCR oligomerization and modulation of transducer coupling properties. Overall, I find the experiments to be technically solid and to include some interesting observations pointing toward D2R dimerization boosting arrestin coupling relative to G protein coupling. However, I do find the experiments relatively indirect, especially the proposed 2:2 D2R:arrestin model, and somewhat over-interpreted. I have some minor suggestions to tone down some key points and to add a little bit more context from the literature:

1. In the introduction, it is worth acknowledging Moller et al, Nat Chem Bio, 2020 as this is an interesting study proposing that arrestin can drive dimerization of the MOR.

We thank the reviewer for the suggestion and have now included this paper in the introduction (page 3, lines 57-58)

2. Overall there is not sufficient acknowledgement of the role of GRKs. Experiments could address if there is a difference in phosphorylation between mutants or KO cells could be used to assess if GRK-independent arrestin-coupling is altered. At least the potential for GRKs to be driving the ultimate effects on arrestin should be stated somewhere.

We have now included some discussion around GRKs and the potential influence this could have on arrestin effects reported here. We now acknowledge that differences in GRK phosphorylation patterns across the mutants could be a potential avenue to explore in a future study and that oligomeric complexes could be composed of a complex of D2R/ β arr2 and GRKs (page 23, lines 474-480).

3. In addition, there should be some more discussion of the potential that mutations have an effect on TMD monomer conformational dynamics and that this could explain some of the functional effect.

We agree this is important to acknowledge and have discussed this on page 22.

4. A more direct measurements of receptor/G protein and receptor/arrestin stoichiometry would be ideal. But, I appreciate that this is quite difficult. Perhaps the authors can more clearly acknowledge that this will eventually be needed to substantiate some of the claims.

We acknowledge that it would be interesting to further validate these receptor- G protein and receptor-arrestin complexes in future studies and have noted this in the discussion (page 25).

5. The variability in arrestin orientation across family A GPCR structures warrants a bit

more discussion. While it is nice that two different orientations are accommodated in 2:2 models, it would be good to assess all known GPCR/arrestin angles and to see if there are any that provide clashes. It is worth noting that tail-only conformations may also contribute which could mitigate this issue.

We thank the Reviewer for raising this relevant point. Indeed, co-author F. Fanelli recently published a study on the structural plasticity of GPCR- β arr1 complexes inferred from the analysis of the available cryoEM structures and Molecular Dynamics simulations (DOI: 10.1016/j.ijbiomac.2024.137217). Regretfully, that study was not mentioned by Marx et al, (bioRxiv, 2025).

In the revised text, we have added the following (see page 19-20): "The analysis of the available cryoEM structural complexes between GPCRs and β arr1 highlighted their high structural plasticity, which was essentially accounted for by the inclination of arrestin with respect to the receptor main axis (Tilt index) and the rotation of arrestin parallel to the membrane plane (Rot index) (Felline et al. Int. J. of Biological Macromolecules, 283:137217, 2024, DOI: 10.1016/j.ijbiomac.2024.137217). According to the Tilt and Rot indices, both the predicted complexes between D2R and β arr2 resemble more the complexes between β arr1 and the β 1-adrenergic receptor (β 1-AR) or the 5HT2B-serotonin receptor (5HT2BR) than the complexes with the other GPCRs (i.e., the m2-muscarinic receptor (M2R, PDB: 6U1N), the NTS1 neurotensin receptor (NTS1R, PDB: 6PWC), the V2-vasopressin receptor (V2R, PDB: 7R0C), and the CB1-cannabinoid receptor (CB1R, PDB: 8WRZ)). By overlapping the M2R, NTS1R, V2R, and CB1R in complex with β arr1 with both D2R protomers in the predicted dimer, no clash between the arrestin molecules was observed. This suggests that the H1-H2 architecture of the D2R dimer in complex with β arr in a 2:2 stoichiometry is compatible with all combinations of Tilt and Rot values explored by the cryoEM GPCR- β arr1 complexes."

6. Along these lines, 2:2 GPCR/arrestin stoichiometries have been recently seen for family C GPCRs, mGluR3 and mGluR8 (et al, Nat Chem Bio, 2025; Marx et al, bioRxiv, 2025) and warrant a mention although it has not been established how the presence of 2 arrestins alters internalization or signaling properties.

Please also see above. We have now cited these studies and an additional study on the adhesion GPCR ADGRE1 (page 18-19)